

# Assessment of a Takagi–Sugeno-Kang fuzzy model assembly for examination of polyphasic loglinear allometry

Hector A. Echavarria-Heras[1], Juan R. Castro-Rodriguez[2], Cecilia Leal-Ramirez[1] and Enrique Villa-Diharce[3]

[1] Departamento de Ecología, Centro de Investigación Científica y de Estudios Superiores de Ensenada, Ensenada, Baja California, México
[2] Facultad de Ciencias Químicas e Ingeniería, Universidad Autónoma de Baja California, Tijuana, Baja California, México
[3] Departamento de Estadística Aplicada, Centro de Investigacion en Matematicas, Guanajuato, Guanajuato, México

## ABSTRACT

**Background**. The traditional allometric analysis relies on log- transformation to contemplate linear regression in geometrical space then retransforming to get Huxley's model of simple allometry. Views assert this induces bias endorsing multi-parameter complex allometry forms and nonlinear regression in arithmetical scales. Defenders of traditional approach deem it necessary since generally organismal growth is essentially multiplicative. Then keeping allometry as originally envisioned by Huxley requires a paradigm of polyphasic loglinear allometry. A Takagi-Sugeno-Kang fuzzy model assembles a mixture of weighted sub models. This allows direct identification of break points for transition between phases. Then, this paradigm is seamlessly appropriate for efficient allometric examination of polyphasic loglinear allometry patterns. Here, we explore its suitability.

**Methods**. Present fuzzy model embraces firing strength weights from Gaussian membership functions and linear consequents. Weights are identified by subtractive clustering and consequents through recursive least squares or maximum likelihood. Intersection of firing strength factors set criterion to estimate breakpoints. A multi-parameter complex allometry model follows by adapting firing strengths by composite membership functions and linear consequents in arithmetical space.

**Results**. Takagi-Sugeno-Kang surrogates adapted complexity depending on analyzed data set. Retransformation results conveyed reproducibility strength of similar proxies identified in arithmetical space. Breakpoints were straightforwardly identified. Retransformed form implies complex allometry as a generalization of Huxley's power model involving covariate depending parameters. Huxley reported a breakpoint in the log–log plot of chela mass vs. body mass of fiddler crabs (*Uca pugnax*), attributed to a sudden change in relative growth of the chela approximately when crabs reach sexual maturity. G.C. Packard implied this breakpoint as putative. However, according to present fuzzy methods existence of a break point in Huxley's data could be validated.

**Conclusions**. Offered scheme bears reliable analysis of zero intercept allometries based on geometrical space protocols. Endorsed affine structure accommodates either polyphasic or simple allometry if whatever turns required. Interpretation of break points characterizing heterogeneity is intuitive. Analysis can be achieved in an interactive way. This could not have been obtained by relying on customary approaches.

Corresponding author
Hector A. Echavarria-Heras, heheras@icloud.com

Besides, identification of break points in arithmetical scale is straightforward. Present Takagi-Sugeno-Kang arrangement offers a way to overcome the controversy between a school considering a log-transformation necessary and their critics claiming that consistent results can be only obtained through complex allometry models fitted by direct nonlinear regression in the original scales.

## INTRODUCTION

Julian Huxley introduced the theory of constant relative growth between a trait $y$ and overall body size $x$ (*Huxley, 1924*; *Huxley, 1932*; *Strauss & Huxley, 1993*). This paradigm is commonly refereed as Huxley's model of simple allometry and is essentially formulated through the power law $y = \beta x^\alpha$ with $\alpha$ identified as the allometric exponent and $\beta$ as the normalization constant. In biology allometric relationships are to within species, as well as, between species (evolutionary allometry) (*Houle et al., 2011*; *Marquet et al., 2005*; *West & Brown, 2005*; *Pélabon et al., 2014*). Power function models are also extensively used in other research fields, e.g., physics (*Newman, 2007*), ecology (*Harris, Duarte & Nixon, 2006*; *Hood, 2007*) earth and atmospheric sciences (*Hills, 2013*), and economics (*Li et al., 2015*). This has encouraged many research endeavors addressing interpretation of involved parameters, as well as, suitability of analysis method for getting estimates. Indeed, concomitant to Huxley's theory of relative growth is the Traditional Analysis Method of Allometry (TAMA hereafter). This is, a widespread device to acquire estimates of the parameters $\alpha$ and $\beta$. It contemplates a logarithmic transformation of the original bivariate data in arithmetical scale in order to consider a linear regression model in geometrical space, and then retransforming to acquire Huxley's model of simple allometry in the original scale. This approach implicitly embraces a notion that variability of the response conforms to a pattern of multiplicative growth. On Huxley's elucidation (*Huxley, 1932*) the intercept $ln\beta$ of TAMA's line was of no specific biological importance, but the slope $b$ was significant enough as to mean allometry itself. This interpretation has permeated contemporary research to such an extent that many practitioners still consider it to be the valid theoretical perspective for static and ontogenetic allometry (*Eberhard, 2009*; *Houle et al., 2011*; *Pélabon et al., 2018*). However, views assert that a TAMA approach produces inconsistent results, thereby recommending allometric examination by relying instead on nonlinear regression in the direct scales of data (*Packard, 2017a*; *Packard, 2013*; *Packard, 2009*; *Packard & Birchard, 2008*). This is, debatable for defenders of the traditional approach that claim that, as it is conceived in the original theoretical context of allometry, a logarithmic transformation deems necessary in the analysis (*Lai et al., 2013*; *Klingenberg, 1998*; *Nevill, Bate & Holder, 2005*; *Kerkhoff & Enquist, 2009*; *Xiao et al., 2011*; *White et al., 2012*; *Ballantyne, 2013*; *Glazier, 2013*; *Niklas & Hammond, 2014*; *Lemaître et al., 2015*; *Pélabon et al., 2018*). Yet steering further away from Huxley's perspective on covariation among different traits, other views conceive allometry centered on the covariation between size and shape (*Mosimann, 1970*; *Klingenberg, 2016*).

From this standpoint, analysis must rely in Multiple Parameter Complex Allometry (MPCA after this) formalizations through all varieties of nonlinear or discontinuous relationships (e.g., *Frankino, Emlen & Shingleton, 2010*; *MacLeod, 2014*; *Bervian, Fontoura & Haimovici, 2006*; *Lovett & Felder, 1989*; *Packard, 2013*). However, adoption of MPCA approaches nourishes one of the most fundamental discrepancies among schools of allometric examination. Indeed, for advocates of the traditional approach, examination based on MPCA models fitted in arithmetical scale sacrifices appreciation of biological theory in order to privilege statistical correctness (*Houle et al., 2011*; *Lemaître et al., 2015*; *Pélabon et al., 2018*). A way to keep the analysis in geometrical space while amending unreliability of a linearity assumption is conceiving the notion of non-log linear allometry (*Packard, 2012b*; *Strauss & Huxley, 1993*; *Echavarría-Heras et al., 2019a*). As every analytic function can be expanded as a power series, curvature in geometrical space has been addressed through polynomial regression schemes (*Kolokotrones et al., 2010*; *Lemaître et al., 2014*; *MacLeod, 2010*; *Glazier, Powell & Deptola, 2013*; *Tidière et al., 2017*; *Echavarría-Heras et al., 2019a*). But besides difficulties related to biological interpretation of a polynomial mean response, this approach maintains a single functional form of the response over the whole covariate range. This could not account for inherent heterogeneity in the logtransformmed response as contemplated in Huxley's theoretical perspective. Certainly, Huxley reported a breakpoint in the log–log plot of chela mass vs. body mass of fiddler crabs (*Uca pugnax*). It was attributed to a sudden change in relative growth of the chela approximately when crabs reach sexual maturity (*Huxley, 1924*; *Huxley, 1927*; *Huxley, 1932*). This suggests a slant aimed at adding complexity in geometrical space while keeping the theoretical essence of traditional allometry in the analytical set up. Is this conception that hosts polyphasic loglinear allometry approaches (PLA afterwards) (*Packard, 2016*; *Gerber, Eble & Neige, 2008*; *Strauss & Huxley, 1993*; *Hartnoll, 1978*). PLA characterizes heterogeneity of the logtransformmed response by composing covariate range into sectors separated by break points. Each subdivision associates to a linear sub model. Broken-line regression (*Beckman & Cook, 1979*; *Ertel & Fowlkes, 1976*; *Tsuboi et al., 2018*; *Ramírez-Ramírez et al., 2019*; *Muggeo, 2003*; *Echavarria-Heras et al., 2019b*). *Forbes & López (1989)* furnish an empirical approach to identification of PLA patterns. Nevertheless, by relying in nonlinear regression this technique requires starting values for the break-point estimation. Therefore, complications set by local maxima, as well as, inferences on estimates could make implementation difficult (*Julious, 2001*; *Muggeo, 2003*).

The quest for new tools that increase reliability of analytical methods has been always a motivation in research. This drive explains the introduction of hybrid models that merge different techniques with the aim of efficiently addressing complexity (*Kimmins, Mailly & Seely, 1999*; *Alur et al., 1995*; *Ajili & Wallace, 2004*; *Pozna et al., 2010*; *Hamilton, Lloyd & Flores, 2017*). In particular, soft computing techniques entail modelling procedures, which are supplemental to customary statistics and probability approaches and that bear tolerance to imprecision, uncertainty, partial truth and approximation (*Baldwin, Martin & Azvine, 1998*). For instance, identification and control of nonlinear systems exemplifies a subject that has greatly benefited by adoption of related hybrid modeling schemes (*Bonissone et al., 1999*; *Kawaji, 2002*; *Vrkalovic, Lunca & Borlea, 2018*; *Chen, 2001*; *Echavarria-Heras et al.,*

*2019b*). Implementation of soft computing protocols include techniques of fuzzy set theory, neural networks, probabilistic reasoning, rough sets, machine learning, and evolutionary computing (*Zadeh, 1993*; *Oduguwa, Tiwari & Roy, 2005*; *Bello & Verdegay, 2012*; *Ibrahim, 2016*; *Al-Kaysi et al., 2017*; *Herrera-Viedma & López-Herrera, 2010*). In this upsurge of nonconventional analytical tools, we can place adaptation of fuzzy logic procedures aimed to lessen parametric uncertainty effects in allometry (*Schreer, 1997*; *Schwetter & Bertone, 2018*; *Bitar, Campos & Freitas, 2016*; *Echavarría-Heras et al., 2018a*; *Näther & Wälder, 2006*; *Dechnik-Vázquez et al., 2019*).

An operating regime based modeling approach offers a structure supporting model adaptation amid an empirical and mechanistic standpoint. Local models valid over restricted domains are combined by smooth interpolation into an overall general output (*Johansen & Foss, 1997*). Therefore, this structure naturally hosts heterogeneity as conceived by PLA (*Echavarria-Heras et al., 2019b*). One example of a hybrid-operating regime based modeling is the Takagi-Sugeno-Kang fuzzy model (*Sugeno & Kang, 1988*; *Takagi & Sugeno, 1985*) (TSK in what follows). This construct composes a fuzzy logic step intended to characterize smoothing weight factors. Then, conventional statistical methods are used to acquire estimates of parameters characterizing sub models. It turns out that the general output of a first order TSK fuzzy model can uniformly approximate any continuous function to arbitrarily high precision (*Ying, 1998*; *Zeng, Nai-Yao & Wen-Li, 2000*). As we show in this examination, an advantage of TSK over conventional PLA, is that it can offer convenient non-statistical proxies of break points for transition among phases. Moreover, consideration of sub models of a TSK scheme as TAMA's linear functions in geometrical space not only offers a congruent PLA model, but it could also entail a highly biologically meaningful model of allometry, because it can model the breakpoints while keeping the meanings of allometric exponents as in Huxley's original formulation. A comprehensive exploration of suitability of the TSK scheme to examine PLA patterns has not been undertaken so here we attempted to fill this gap. In what follows a formulation of PLA by means of a the TSK fuzzy model will be referred as TSK-PLA for short.

The outstanding approximation capabilities of a TSK fuzzy model entail reliable identification of whatever MPCA functional form renders necessary in arithmetical space (*Echavarría-Heras et al., 2018a*; *Echavarría-Heras et al., 2019b*). Adaptation of the TSK fuzzy model for that aim will be forward designated by means of the TSK-MPCA abbreviation. As a criterion to evaluate the performance of the TSK-PLA proxy we verified the dependability of linked retransformation results, including break point placement and reproducibility strength of mean response function against corresponding estimations produced via TSK-MPCA. It turns out that proposed TSK-PLA analysis method endorsed reliable identification of heterogeneity of examined allometries. Furthermore, the affine structure of the present fuzzy protocol can accommodate either complex or simple allometry as required to analyzing the data. Thus, the presented TSK-PLA model can be considered as a general tool for examination of zero intercept allometries. Moreover, from a theoretical standpoint a TSK-PLA representation implies an allometric model in arithmetical space that seemingly fits MPCA. This expresses the response as a generalized power function including scaling parameters expressed as functions of the covariate

(*Bervian, Fontoura & Haimovici, 2006*; *Echavarría-Heras et al., 2019a*; *Echavarria-Heras et al., 2019b*). But, above all, present fuzzy approach contributes by offering a way of overcoming the controversy between a school considering analysis in geometrical space as a must in allometry, and critics claiming that consistent results can only come along by using a MPCA formulation followed by nonlinear regression protocol in the original scale of data. Interestingly, present TSK-PLA arrangement also contributed on qualitative grounds. Certainly, Huxley reported a breakpoint in the log–log plot of chela mass vs. body mass of fiddler crabs (*Uca pugnax*). *Packard (2012a)* inferred this point was only putative. In his own interpretation, perhaps due to combined effects of a log transformation itself and the format of graphical display of Huxley's data. However, application of present Takagi-Sugeno-Kang protocol supports existence of a break point in Huxley's Uca *pugnax* log–log plot.

This article is organized as follows: In the Materials and Methods section, we formally explain the steps backing the identification of the offered TSK-PLA scheme. There, we explain why this construct can be considered as a generalized protocol for allometric analysis in geometrical space. We also clarify why the offered TSK model implies a MPCA scheme in arithmetical space. The presentation includes an elucidation of sufficient conditions under what the asymptotic mode of the acquired TSK proxy behaves as the power function in Huxley's model of simple allometry. There, we also suggest a correction factor (CF here after) for bias of retransformation of the regression error that grants highest reproducibility for derived mean response function in arithmetical space. The Results section highlights on the advantages of the present approach over conventional counterparts. A Discussion section elaborates on the contribution that our approach bears for the general subject of suitability of analysis method in bivariate allometry. An Appendix includes a detailed explanation of the steps involved in the construction and identification of the general form of the addressed TSK models.

## MATERIALS & METHODS

### Data

Allometric examination here mainly relied on a primary data set exhibiting curvature in geometrical space. This composes 10,412 measurements of *Zostera marina* (Eelgrass) leaf biomasses *y* and corresponding leaf areas *x* as reported in *Echavarría-Heras et al. (2019a)* and *Echavarría-Heras et al. (2018b)*. For comparison, we also considered data reported in *Mascaro et al. (2011)* comprising 30 Biomass-Diameter at Breast Height measurements on *Metrosideros polymorpha*. Analisis also extended to data reported in *De Robertis & Williams (2008)* including 29,363 Length–Weight measurements on *Gadus chalcogrammu*. This last data set allowed illustration of the performance of the TSK paradigm in a circumstance where the TAMA protocol is consistent. Finally, we analysed the fitness of the TSK in detecting break points in the log–log plot of chela mass vs. body mass of fiddler crabs (*Uca pugnax*) (*Huxley, 1924*; *Huxley, 1932*).

## Models
### General formula of allometry

The methods engaged here aim to identification of the suitable form of the allometric function representing the variation of a trait $y$ depending on a descriptor $x$. For that purpose, we firstly introduce the formal framework and the notation convention used through. We assume that a response $y$ and its covariate $x$ belong to domains $Y$ and $X$ of positive real numbers one to one and with $y$ having a zero limit when $x$ approaches zero. We also consider that there exist a function $w(x, \boldsymbol{p}) : X \rightarrow Y$ where $\boldsymbol{p} = (p_1, \dots, p_n)$ is a parameter set, and a concomitant approximation error function $\epsilon(x) : X \rightarrow Y$ that combine to model whatever form, the linkage between $x$ and $y$ acquires. Moreover, we take on, that such a relationship can be expressed through an additive error description

$$y = w(x, \boldsymbol{p}) + \epsilon(x) \tag{1}$$

or else through the multiplicative error alternate

$$y = w(x, \boldsymbol{p}) e^{\epsilon(x)} \tag{2}$$

In order to get $w(x, \boldsymbol{p})$, we can consider the error term $\epsilon(x)$ as a random variable $\epsilon$. Then, specifications above offer two commonly addressed analysis protocols in allometry. A regression model with additive error in arithmetical scale

$$y = w(x, \boldsymbol{p}) + \epsilon \tag{3}$$

with $\epsilon$ taken as $\psi-$distributed with zero mean and variance generally expressed as a function $\sigma^2(x)$ of covariate, that is, $\epsilon \sim \psi(0, \sigma^2(x))$. Fitting Eq. (3) generally requires direct nonlinear regression protocols. This returns a mean response function $E_{aw}(y|x) = w(x, \boldsymbol{p})$. For the sake of facilitating comparison aims in further developments, this $a$ subscript will be maintained to typify a mean response function gotten by means of identification protocols applied in arithmetical space.

A second procedure circumscribes to the multiplicative error model of Eq. (2) and relies in a logtransformation procedure in order to consider a parallel regression model in geometrical space. Formally, we contemplate a mapping $(y, x) \rightarrow (v, u)$ such that $u = lnx$ and $v = lny$. This sets variation domains $U$ and $V$ for $u$ and $v$ to one. We concomitantly have the regression model with additive error in geometrical space

$$v = v(u, \boldsymbol{\pi}) + \epsilon, \tag{4}$$

where formally

$$v(u, \boldsymbol{\pi}) = ln(w(x, \boldsymbol{p})) \tag{5}$$

and $\epsilon$ is random variable as specified above. It follows that back-transforming Eq. (4) to arithmetical space yields,

$$y = \exp(v(u, \boldsymbol{\pi})) e^{\epsilon}. \tag{6}$$

Then, concomitant mean response function is symbolized through $E_{gw}(y|x)$ and becomes

$$E_{gw}(y|x) = \exp(v(u, \boldsymbol{\pi})) \delta \tag{7}$$
where $\delta = E(e^\epsilon)$. Notice that in $E_{gw}(y|x)$ we have used the notation convention of a subscript $g$ referring to identification of $w(x, \boldsymbol{p})$ based on the regression model of Eqs. (4) and (5).

The CF, $\delta$ above provides the necessary adjustment for bias of retransformation of the regression error $\epsilon$ (*Mascaro et al., 2011*; *Baskerville, 1972*; *Newman, 1993*). Assuming $\epsilon \sim N\left(0, \sigma^2\right)$ sets $e^\epsilon$ to be lognormally distributed. Then, CF becomes

$$\delta = e^{\sigma^2/2}. \tag{8}$$

But, *Newman (1993)* asserts that whenever $\epsilon$ is not normally distributed, $\delta$ is given by the smearing estimate of bias of *Duan (1983)*. Nevertheless, in some settings this nonparametric form can produce bias overcompensation (*Manning, 1998*; *Smith, 1993*; *Koch & Smillie, 1986*). *Zeng & Tang (2011a)* propose an alternate nonparametric form of $\delta$ namely

$$\delta = 1 + \sigma^2/2. \tag{9}$$

Actually, $\delta$ given this way corresponds to a three terms partial sum approximation of the power series expression of $E(e^\epsilon)$ assuming $E(\epsilon) = 0$. By the same token, *Echavarría-Heras et al. (2019a)* suggest a representation for $\delta$ given by a $n$-terms partial sum of series representation of $E(e^\epsilon)$, that is,

$$\delta = \sum_{0}^{n} \frac{E(\epsilon^k)}{k!}. \tag{10}$$

Maximization of Lin's Concordance Correlation Coefficient (CCC) (*Lin, 1989*) between observed values and mean response $E_g(y|x)$ resulting using this form of $\delta$ sets criterion to choose $n$.

### Huxley's formula of Simple Allometry

A characterization of $w(x, \boldsymbol{p})$ as a power function $\beta x^\alpha$ has been traditionally referred as Huxley's formula of simple allometry (*Strauss & Huxley, 1993*). This model will be ahead epitomized by a subscript $s$ as a mnemonic device for "simple". Equation (3) becomes

$$y = w_s(x, \boldsymbol{p}) + \epsilon \tag{11}$$

with $w_s(x, \boldsymbol{p}) = \beta x^\alpha$ and $\epsilon$ assumed to be normally distributed with zero mean and variance $\sigma^2$, that is, $\epsilon \sim \psi\left(0, \sigma^2\right)$. According to our notation convention Eq. (11) yields the mean response function $E_{as}(y|x) = \beta x^\alpha$.

Similarly, the logtransformation method produces the TAMA's regression model, that is,

$$v = v_s(u, \boldsymbol{\pi}) + \epsilon \tag{12}$$

with

$$v_s(u, \boldsymbol{\pi}) = ln\beta + \alpha u \tag{13}$$

and $\epsilon$ as specified above. Equations (12) and (13) determine $E_s(v|u) = v_s(u, \boldsymbol{\pi})$. Accordingly, back transformation of Eq. (12) to arithmetical space brings up a mean response $E_{gs}(y|x)$ given by

$$E_{gs}(y|x) = \beta x^\alpha \delta, \tag{14}$$

where $\delta$ stands for necessary CF.

It often occurs that even after contemplation of proper form for $\delta$ this TAMA's derived proxy for $w(x, \boldsymbol{p})$ produces biased projections of observed values of the response $y$. This means, that complexity of Huxley's formula of simple allometry $w_s(x, \boldsymbol{p})$ becomes inappropriate to identify the true $w(x, \boldsymbol{p})$ form (*Gould, 1966*; *Huxley, 1932*; *Bervian, Fontoura & Haimovici, 2006*; *MacLeod, 2014*; *Echavarría-Heras et al., 2019a*). From the settings of Eq. (1) it is reasonable assuming that adapting complexity as it is needed to identify $w(x, \boldsymbol{p})$ could depend on MPCA forms. Corresponding logtransformmed expression $v(u, \boldsymbol{\pi})$ is inferred to be a nonlinear function of covariate $u$. This rears PLA as a likely device to acquire complexity for identification of MPCA through geometrical space methods. We adopt the affine structure of a first order TSK fuzzy model as a device for identification both MPCA or PLA alternates.

### The TSK account of $w(x, \boldsymbol{p})$

The general output of a first order TSK fuzzy model bears a fuzzy alternate to a statistical mixture regression model (*Cohn, Ghahramani & Jordan, 1997*). It is then reasonable to assume that such an structure could efficiently address the problem of identifying $w(x, \boldsymbol{p})$ expressed as a MPCA model in arithmetical scale or its assumed PLA forms in geometrical space. The symbol $w_{TSK}(x, \boldsymbol{p})$ will stand for the TSK-MPCA surrogate for $w(x, \boldsymbol{p})$. Accordingly, adaptation of Eq. (3) becomes

$$y = w_{TSK}(x, \boldsymbol{p}) + \epsilon_{TSK} \tag{15}$$

with $\epsilon_{TSK}$ a $\psi-$distributed residual random variable with zero mean and variance expressed as a function $\sigma^2_{TSK}(x)$ of $x$, that is, $\epsilon_{TSK} \sim \psi(0, \sigma^2_{TSK}(x))$.

Denoting through the symbol $E_{aTSK}(y|x)$ the mean response function determined by Eq. (15), we have

$$E_{aTSK}(y|x) = w_{TSK}(x, \boldsymbol{p}). \tag{16}$$

Since, the general output of a first order TSK fuzzy model can uniformly approximate any continuous function to arbitrarily high precision (*Ying, 1998*; *Zeng, Nai-Yao & Wen-Li, 2000*) then whatever MPCA form $w(x, \boldsymbol{p})$ embraces, this can be accurately projected through a consistent identification of $E_{aTSK}(y|x)$.

In turn, according to Eq. (4) the TSK-PLA regression protocol becomes,

$$v = v_{TSK}(u, \boldsymbol{\pi}) + \epsilon_{TSK} \tag{17}$$

where according to Eq. (5), $v_{TSK}(u, \boldsymbol{\pi}) = \ln(w_{TSK}(x, \boldsymbol{p}))$ and $\epsilon_{TSK}$ as specified in Eq. (15).

In turn, equation Eq. (17) yields $E_{TSK}(v|u) = v_{TSK}(u, \boldsymbol{\pi})$. Additionally, a backtransformation $e^v$ of Eq. (17) sets

$$y = \exp(v_{TSK}(u, \boldsymbol{\pi}))e^{\epsilon_{TSK}}. \tag{18}$$

Then, corresponding mean response function in arithmetical space turns out to be

$$E_{gTSK}(y|x) = \exp(v_{TSK}(u, \boldsymbol{\pi}))\delta. \tag{19}$$

By assumption, we take $v_{TSK}(u, \boldsymbol{\pi})$ in the form given by Eq. (A14), that is,

$$v_{TSK}(u, \boldsymbol{\pi}) = \sum_{1}^{q} \vartheta^i(u) f^i(u) \qquad (20)$$

with $\vartheta^i(u)$ and $f^i(u)$ a one to one the firing strengths and consequent functions for $i = 1, 2, \ldots, q$. Since, the domain $U$ of covariate is $R$, we can assume membership functions $\mu_{\Phi_k}(u)$ to have a Gaussian form i.e.,

$$\mu_{\Phi_k}(u) = exp\left\{ -\frac{1}{2}\left[ \left(\frac{u - \theta_k}{\lambda_k}\right)^2 \right] \right\} \qquad (21)$$

being $\theta_k$ and $\lambda_k$ for $k = 1, 2, \ldots, q$, parameters. We also consider that consequent local models $f^i(u)$ have a description, that is,

$$f^i(u) = ln\beta_i + \alpha_i u \qquad (22)$$

being $\alpha_i$ and $ln\beta_i$ parameters. Readily, Eqs. (20) through (22) entail a TSK-PLA arrangement. As it will be clarified ahead a similar adaptation of Eq. (A14) stablishes the TSK-MPCA form $w_{TSK}(x, \boldsymbol{p})$ in Eq. (15).

Identification of $v_{TSK}(u, \boldsymbol{\pi})$ as given by Eqs. (20) through (22) is performed by means of the Matlab function: **main_fun_tsk_pla_model_fit.m**. available from the Supplementary Information section. Heterogeneity and reproducibility strength features of $v_{TSK}(u, \boldsymbol{\pi})$ can be interactively explored through different characterizations of the clustering radius-training parameter $r_a$ as specified by Eqs. (B7) through (B9).

As described in Appendix A, acquiring $v_{TSK}(u, \boldsymbol{\pi})$ requires on first stage a fuzzy partition $L_u$ of the input domain $U$ (cf. Eq. (A3)). Achieving this relies on a Subtractive Clustering (SC after this) technique to establish the value of the parameter $q$ (*Castro et al., 2016*; *Chiu, 1994*). A brief description of the SC method is provided in Appendix B. This stage also sets the number of inference rules $R^i$ specified by Eq. (A10) and concomitant number local models in $v_{TSK}(u, \boldsymbol{\pi})$. The SC step also produces estimates for the parameters $\theta_k$ and $\lambda_k$ in characterizing the membership functions $\mu_{\Phi_k}(u)$. Then, the normalized firing strength functions, $\vartheta^i(u)$ follows from (Eqs. A11) and (A12). A second step targets at characterization of the linear consequents $f^i(u)$ as given by Eq. (22). This is achieved by replacing the identified factors $\vartheta^i(u)$ and the assumed form of the consequents $f^i(u)$ into Eq. (20) to characterize the regression model of Eq. (17). Then, the parameters in the consequents $f^i(u)$ are estimated by implementing a Recursive Least Squares (RLS) routine (*Jang, Sun & Mizutani, 1997*; *Wang & Mendel, 1992*). This identification step could be also performed through a maximum likelihood approach (*Kalbfleisch, 1985*). The whole identification algorithm is explained in Appendix B. Codes are included in the Supplementary Information section.

### Assessment of reproducibility strength of models

Following *Echavarria-Heras et al. (2019b)* reproducibility will be primarily estimated by comparing values of Lin's concordance correlation coefficient, symbolized by means of $\boldsymbol{\rho_C}$ (*Lin, 1989*). Agreement will be defined as poor whenever $\boldsymbol{\rho_C} < 0.90$, moderate for

$0.90 \leq \rho_C < 0.95$, good for $0.95 \leq \rho_C < 0.99$, or excellent for $\rho_C \geq 0.99$ (*McBride, 2005*). Assessment of reproducibility will also rely on model performance metrics, such as the Coefficient of Determination (CD), Standard Error of Estimate (SEE), Mean Prediction Error (MPE), and Mean Percent Standard Error (MPSE) (*Gupta, Sorooshian & Yapo, 1998*; *Hauduc et al., 2011*; *Zeng et al., 2017*; *Zeng & Tang, 2011b*; *Parresol, 1999*; *Meyer, 1938*; *Schlaegen, 1982*). Related statistics are included in Appendix C. Matlab and R codes are provided in the supplemental files section.

## RESULTS

### Data

Plots in Fig. 1 display the spread response–covariate in geometrical space for data sets included in this examination. Figure 1A relates to the *Echavarría-Heras et al. (2019a)* data. Figure 1B is for *Mascaro et al. (2014)* data. Figure 1C shows Huxley's plot of chela mass vs. body mass of fiddler crabs (*Uca pugnax*) in log–log scale (*Huxley, 1924*; *Huxley, 1932*). Figure 1D displays spread for the *De Robertis & Williams (2008)* data. Assessment of curvature will be performed for all data sets by analyzing fitting results of the TSK-PLA and TSK-MPCA models. For easy of presentation detailed results on a TAMA-TSK model comparison will only circumscribe to the *Echavarría-Heras et al. (2019a)* data.

### Representation of the back-transformed TSK-PLA proxy as a MPCA formula

This section explains that assuming TSK-PLA implies a multiple parameter complex allometry form in direct arithmetical scales. Indeed following *Bervian, Fontoura & Haimovici (2006)*, *Echavarría-Heras et al. (2019a)* proposed an extension of Huxley's formula of simple allometry $w_s(a, \boldsymbol{p}) = \beta a^\alpha$ that includes scaling parameters $\alpha$ and $\beta$ depending in the covariate, that is,

$$y = \beta(x) x^{\alpha(x)} \tag{23}$$

where $\beta(x)$ and $\alpha(x)$ are continuous functions and with $\beta(x)$ assumed to be positive. This sets

$$w(x, \boldsymbol{p}) = \beta(x) x^{\alpha(x)}. \tag{24}$$

Thus, formally whenever the scaling functions $\beta(x)$ and $\alpha(x)$ are not simultaneously constant $w(x, \boldsymbol{p})$ as given by Eq. (24) entails MPCA (*Echavarría-Heras et al., 2019a*; *Echavarria-Heras et al., 2019b*).

We now demonstrate that the mean response function $E_{gTSK}(y|x)$ in arithmetical space derived from a TSK-PLA arrangement implies MPCA in the form set by Eq. (24). For that drive, we notice that replacing Eq. (22) into Eq. (20) and then rearranging leads to

$$v_{TSK}(u, \boldsymbol{\pi}) = ln\beta_{TSK}(u) + \alpha_{TSK}(u) u \tag{25}$$

where

$$\beta_{TSK}(u) = e^{\sum_1^q \vartheta^i(u) ln\beta_i} \tag{26}$$

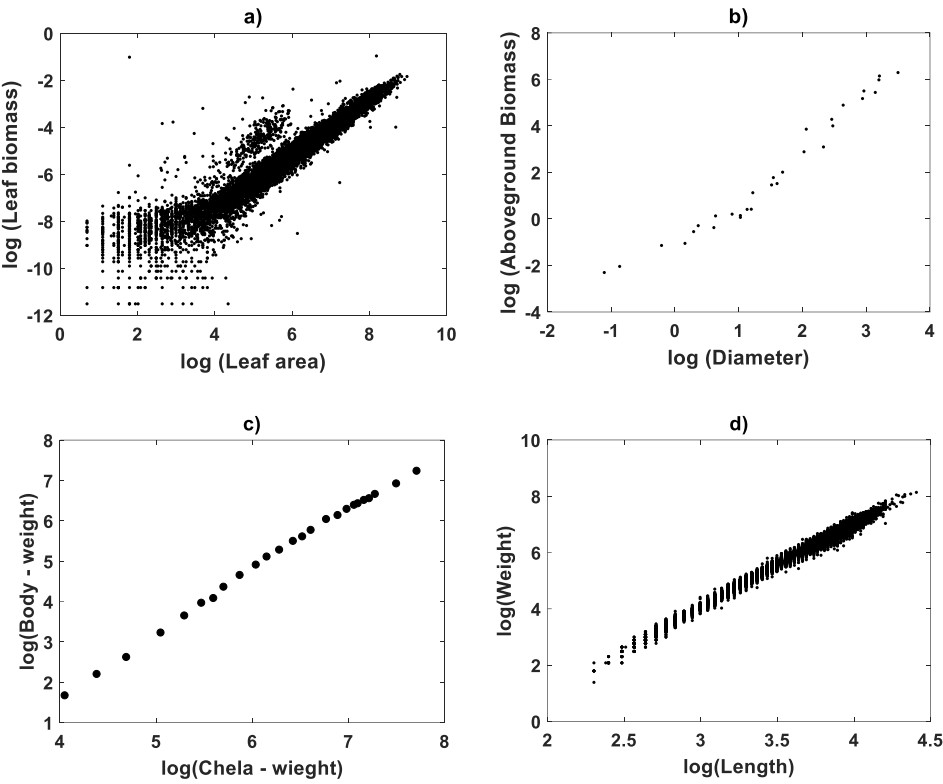

**Figure 1 Spreads of allometric response and covariate in geometrical space.** This plot shows the spread response–covariate in geometrical space for the included data sets. (A) depicts dispersion for *Echavarría-Heras et al. (2019a)*, (B) presents that associating to *Mascaro et al. (2014)* (C) shows that for *Huxley (1932)* and (D) is for the *De Robertis & Williams (2008)* data sets.

and

$$\alpha_{TSK}(u) = \sum_{1}^{q} \vartheta^i(u)\alpha_i. \tag{27}$$

Thus, Eq. (17) takes on the equivalent representation,

$$v = ln\beta_{TSK}(u) + \alpha_{TSK}(u)u + \epsilon_{TSK}. \tag{28}$$

The functions $ln\beta_{TSK}(u)$ and $\alpha_{TSK}(u)$ above, suggest $u-$ dependent forms of the parameters $ln\beta$ and $\alpha$ involved in the regression model of the TAMA approach. Then, under the assumption of Eq. (22) a TSK–PLA arrangement interprets as generalization of the TAMA scheme (*Echavarría-Heras et al., 2019a*). Applying the back-transformation $e^v$ of Eq. (28) and recalling Eq. (19) yields

$$E_{gTSK}(y|x) = \beta_{TSK}(x)x^{\alpha_{TSK}(x)}\delta. \tag{29}$$

This sets $exp(v_{TSK}(u,\boldsymbol{\pi})) = \beta_{TSK}(x)x^{\alpha_{TSK}(x)}$. But, from Eq. (5) we have $v_{TSK}(x,\boldsymbol{p}) = ln(w_{TSK}(x,\boldsymbol{p}))$ then $w(x,\boldsymbol{p})$ as identified by retransformation of $v_{TSK}(x,\boldsymbol{p})$ admits the form specified by Eq. (24).

**Table 1  Fitting results of the TAMA protocol for the *Echavarría-Heras et al. (2019a)* data set.**

**Residual statistics**

| Minimum | 1Q | Median | 3Q | Maximum |
|---|---|---|---|---|
| −4.7535 | −0.2642 | 0.0042 | 0.2151 | 8.3509 |

**Coefficient values**

| Parameters | Estimate | Std. Error | t value | Pr(>|t|) | Confidence Interval (95%) |
|---|---|---|---|---|---|
| $\alpha$ | 1.022775 | 0.003662 | 279.3 | <2e−16 | (1.015597, 1.029953) |
| $ln\beta$ | −11.202199 | 0.021515 | −520.7 | <2e−16 | (−11.24437, −11.16003) |

**Fitting test.**

| Test | | | Value | | |
|---|---|---|---|---|---|
| Residual standard error | | | 0.5723 on 10410 degrees of freedom | | |
| Multiple R-squared | | | 0.8823 | | |
| Adjusted R-squared | | | 0.8823 | | |
| F-statistic | | | 7.802e+04 on 1 and 10410 DF | | |
| *p*-value | | | <2.2e−16 | | |

### Fitting results of the TAMA protocol: *Zostera marina*

For comparison aims, we present fitting results of the TAMA on the *Echavarría-Heras et al. (2019a)* data. Estimates for the allometric parameters $\alpha$ and $\beta$ derive from linear regression on log-transformed data $(v, u)$ (cf. Eq. (12)). Table 1 summarizes the results of the analysis. Corresponding, mean response $E_s(v|u)$ in geometrical scale is shown in Fig. 2A. Log-transformation is a mechanism aimed to reduce variability of data (*Feng et al., 2014*). Nevertheless, Fig. 2A still displays a significant dispersion of $v$ values about $E_s(v|u)$. Spread may lead on first sight to the impression that the distribution of $v$ around the mean response line $E_s(v|u)$ for small values of $u$ is fair. Agreeing with (*Packard, 2017b*), on the importance of graphs in allometry, led to a careful revision of the spread which suggested curvature. Moreover, the assessment of dispersion of residuals $\epsilon$ of Eq. (12) suggested lack of normality, as well as, heteroscedasticity (Fig. 2B). Further, QQ plot shows heavier tails than expected for a normal distribution (Fig. 2C). Indeed, an *Anderson & Darling (1952)* test to ascertain normality of $\epsilon$ residuals produced a test statistics value of A = 310.848 and a : $p$-value < 2.2e–16. In turn a Lilliefors (Kolmogorov-Smirnov-type) test, delivered D = 0.1305, as well as, a relatively small $p$-value < 2.2e–16. Therefore, the hypothesis of normally distributed errors in the analysis should be rejected since obtained $p$-values are extremely small (< 2.2e–16). What is more, a lack of normality of $\epsilon$ errors in the linear regression analysis of Eq. (12) can be also ascertained from the normal QQ plot shown in Fig. 2C. It can be perceived that the distribution of $\epsilon$ residuals exhibits heavier tails than those expected for a normal distribution.

Besides, a Breush-Pagan statistic (*Breusch & Pagan, 1979*), provided a way to assess heteroscedasticity of the $\epsilon$ residuals. In order to perform this test, the squared errors in the linear model of Eq. (12) were assumed to depend linearly on the independent variable i.e.,

$$\epsilon(u)^2 = b + du + \zeta \tag{30}$$

where $b$ and $d$ are parameters and $\zeta$ the error term. The null hypothesis is that the parameter $d$ in Eq. (30) vanishes. Rejection of the null hypothesis not only corroborates

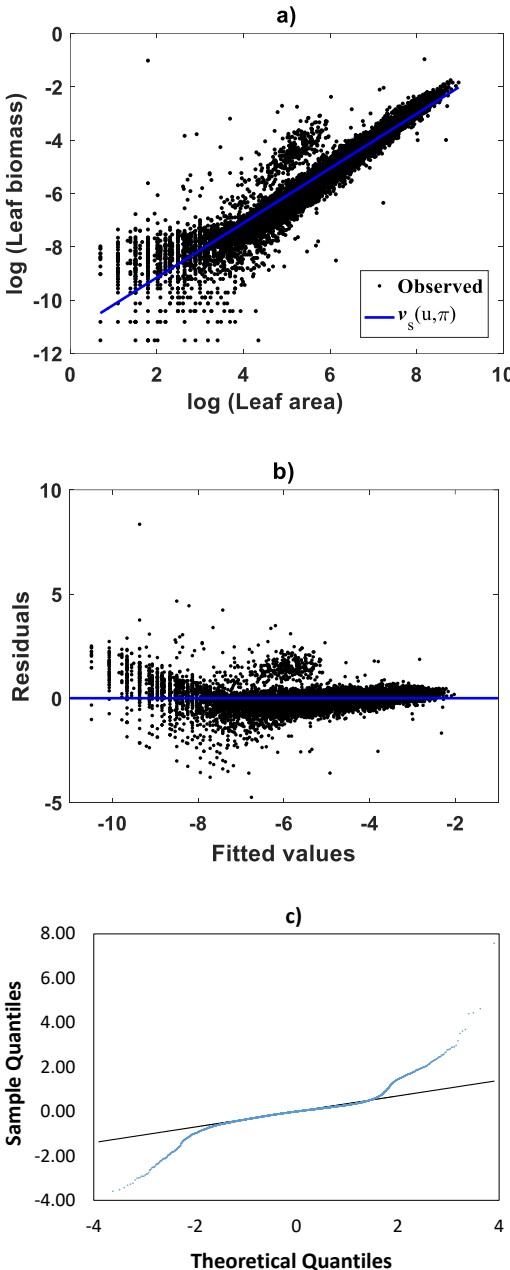

**Figure 2** **Dispersion plots for the TAMA fit on the** *Echavarría-Heras et al. (2019a)* **data set.** (A) shows dispersion of log transformed eelgrass leaf biomasses *v* around the estimated form of mean response line of Eq. (13). Residuals for the regression model of Eq. (12) show irregular spreading about the zero line (B). Besides, the QQ plot in (C) displays heavier tails than expected for a normal distribution.

heteroscedasticity but also provides information on variability. The test statistics turned out to be BP = 808.8119 with one degree of freedom with a $p-$value ($< 2.2e{-}16$), that is sufficiently small as to provide strong evidence against homoscedasticity, while undoubtedly favoring heteroscedasticity. Thus, the presently fitted straight line does not comply the

essential assumptions of normality and homoscedasticity of the analysis. Therefore, the TAMA protocol turned out unsuited for analyzing the allometric relationship in the *Echavarría-Heras et al. (2019a)* data. Consequently, characterization of the general function $w(x, \boldsymbol{p})$ entailed by Huxley's model of simple allometry (cf. Eq. (11)) does not fit required complexity. Thus data spread shown in Fig. 2A, submits curvature in geometrical space. We now turn to explore the capabilities of the TSK-PLA construct to identify this pattern.

### Fitting results of the TSK-PLA assembly: *Zostera marina*
#### Identification of firing strength factors $\vartheta^i(u)$
In order to identify the required firing strength factors $\boldsymbol{\vartheta^i}(\boldsymbol{u})$ for $i = 1, 2, \ldots, q$. We executed the **main_fun_tsk_pla_model_fit.m** code on log-transformed values $(v, u)$ from the *Echavarría-Heras et al. (2019a)* data set. This try assumed membership functions $\mu_{\Phi_k}(u)$ having a form given by Eq. (21) for $k = 1, 2, \ldots, q$. Setting $r_a = 0.47$ returned a value of $q = 2$. Then, we have to consider two membership functions characterizing the fuzzy partition of imput space $U$. Moreover, in compliance with Eq. (A12) normalized firing strength factors $\vartheta^1(u)$ and $\vartheta^2(u)$ turn out to be

$$\vartheta^1(u) = \cfrac{1}{1 + \exp\left\{-\frac{1}{2}\left[\left(\frac{u-\theta_2}{\lambda_2}\right)^2 - \left(\frac{u-\theta_1}{\lambda_1}\right)^2\right]\right\}} \tag{31}$$

$$\vartheta^2(u) = \cfrac{\exp\left\{-\frac{1}{2}\left[\left(\frac{u-\theta_2}{\lambda_2}\right)^2 - \left(\frac{u-\theta_1}{\lambda_1}\right)^2\right]\right\}}{1 + \exp\left\{-\frac{1}{2}\left[\left(\frac{u-\theta_2}{\lambda_2}\right)^2 - \left(\frac{u-\theta_1}{\lambda_1}\right)^2\right]\right\}}. \tag{32}$$

Plots of the estimated membership functions $\mu_{\Phi_1}(u)$ and $\mu_{\Phi_2}(u)$ and normalized firing strength factors $\vartheta^1(u)$ and $\vartheta^2(u)$ appear in Fig. 3A and Fig. 3B respectively. We observe that agreeing curves intersect at a point $u_b = 3.998$.

#### Identification of consequent functions $f^i(u)$
A second step in the procedure to get $v_{TSK}(u, \pi)$ concerns acquiring the consequent functions $f^i(u)$ in Eq. (22). Since, for this data, we obtained $q = 2$, the code ought to establish consequent functions $f^1(u)$ and $f^2(u)$, each one assumed to be linear, that is,

$$f^1(u) = ln\beta_1 + \alpha_1 u \tag{33}$$

and

$$f^2(u) = ln\beta_2 + \alpha_2 u. \tag{34}$$

With the aim of assessing heteroscedasticity, we replaced the forms of $\vartheta^1(u)$ and $\vartheta^2(u)$ identified by SC technique in regression Eq. (17). In turn the involved consequent functions $f^1(u)$ and $f^2(u)$ were assumed to have both the form given by Eqs. (33) and (34) correspondingly. Then, we assumed the involved $\epsilon_{TSK}$ residuals to be normally distributed with zero mean, but with a standard deviation set as a function $\sigma_{TSK}(u)$ of the covariate $u$, Namely

$$\sigma_{TSK}(u) = \log(\sigma + ku), \tag{35}$$

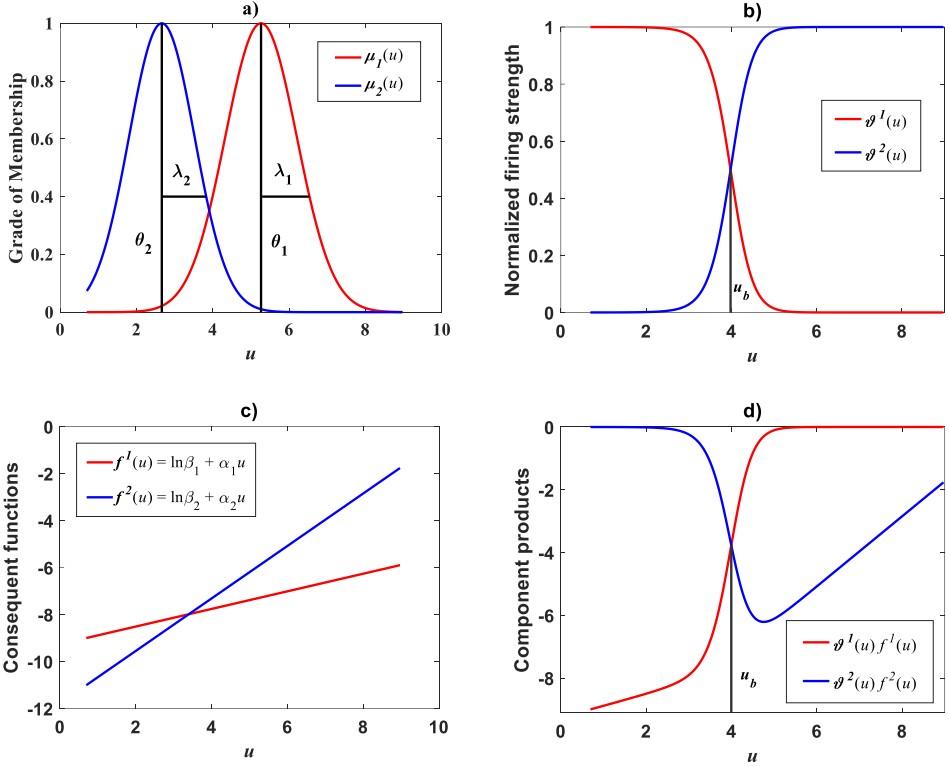

**Figure 3** **Elements of the TSK-PLA fuzzy model identified on the *Echavarría-Heras et al. (2019a)* data set.** Shown results associate to a value $r_a = 0.47$ for the clustering radius that corresponded to a $q = 2$, heterogeneity index. (A) displays plots of membership functions both given in the Gaussian form of Eq. (21). (B) presents plots of normalized firing strength factors given by Eqs. (31) and (32) one to one. A break point at $u_b = 3.98$ is shown. (C) displays consequent linear functions as given by Eqs. (33) and (34). (D) portraits component products.

where $\sigma$ and $k$ are parameters to be estimated and such that $\sigma + ku > 1$. Table 2 presents maximum-likelihood parameter estimates and associated uncertainties for the related fit. We can ascertain a highly significant fit, since, in all cases the standard error is extremely small, this mainly explained by the large amount of data in the analysis. To judge heteroscedasticity of residuals we study the confidence interval of parameter $k$. This parameter determines the change in residual variability per unit change in $u$. It turns out that figures in Table 2 show that confidence interval of parameter $k$ does not include a zero value. Therefore, we may conclude that with high probability the variability of the residuals changes as $u$ changes.

Meanwhile, setting $k = 0$ in Eq. (35) allowed consideration of a parallel maximum likelihood fit of homoscedastic case of the TSK regression model of Eq. (17). Table 3 provides fitting results. Model performance metrics allow assessment of the fits of the heteroscedastic and homoscedastic versions of the TSK–PLA protocol. Accordingly, we can assert that the heteroscedastic model stands a better fit than its homoscedastic counterpart. Particularly, in the heteroscedastic case we have a negative log-likelihood

**Table 2** Fitting results of the TSK-PLA regression model of Eq. (17) for the *Echavarría-Heras et al. (2019a)* data set, assuming heteroscedasticity in the form given by Eq. (35).

**Residual statistics**

| Minimum | 1Q | Median | 3Q | Maximum |
|---|---|---|---|---|
| −4.4478 | −0.2273 | $-5.99 \times 10^{-5}$ | 0.1936 | 7.5718 |

**Fitting results**

| Parameters | Estimate | Std. Error | *t* value | Pr(>|t|) | Confidence Interval (95%) |
|---|---|---|---|---|---|
| $ln\beta_1$ | −11.7672 | 0.0259 | −454.09 | $<2.2 \times 10^{-16}$ | (−11.8180613 −11.7164792) |
| $\alpha_1$ | 1.1148 | 0.0036 | 302.84 | $<4.0 \times 10^{-16}$ | (1.1076198 1.1220501) |
| $ln\beta_2$ | −9.2326 | 0.0803 | −114.93 | $<2.2 \times 10^{-16}$ | (−9.3901440 −9.0752428) |
| $\alpha_2$ | 0.3629 | 0.0278 | 13.02 | $<4.2 \times 10^{-39}$ | (0.3083460 0.4175571) |
| $\sigma$ | 2.5063 | 0.0169 | 147.93 | $<2.2 \times 10^{-16}$ | (2.4731764 2.5395905) |
| $k$ | −0.1580 | 0.0021 | −73.21 | $<2.2 \times 10^{-16}$ | (−0.1623302 −0.1538654) |

**Table 3** Fitting results of the TSK-PLA regression model of Eq. (17) for the *Echavarría-Heras et al. (2019a)* data set, assuming homoscedasticity.

**Residual statistics**

| Minimum | 1Q | Median | 3Q | Maximum |
|---|---|---|---|---|
| −4.4569 | −0.2264 | 0.0017 | 0.1943 | 7.5545 |

**Fitting results**

| Parameter | Estimate | Std. Error | t value | Pr(>|t|) | Confidence Interval (95%) |
|---|---|---|---|---|---|
| $ln\beta_1$ | −11.7029 | 0.0369 | −316.34 | $<2.2 \times 10^{-16}$ | (−11.7754563, −11.6304399) |
| $\alpha_1$ | 1.1047 | 0.0058 | 189.14 | $<2.2 \times 10^{-16}$ | (1.0932778, 1.1161734) |
| $ln\beta_2$ | −9.1869 | 0.0562 | −163.27 | $<2.2 \times 10^{-16}$ | (−9.2972775, −9.0767044) |
| $\alpha_2$ | 0.3437 | 0.0205 | 16.69 | $<7.1 \times 10^{-63}$ | (0.3033585, 0.3840624) |
| $\sigma$ | 0.5273 | 0.0036 | 144.30 | $<2.2 \times 10^{-16}$ | (0.5201765, 0.5345015) |

value of $-logL = 6304.60$, which turns out to be notably smaller than $-logL = 8111.49$ obtained for the homoscedastic model. These figures bear a notable difference that backs the selection of the heteroscedastic model. This difference in fit quality favoring the heteroscedastic model is mainly due to the fact that the latter models the pattern of variation of the errors in a more reliable way. Plots of identified consequents appear in Fig. 3C, component products $\vartheta^1(u)f^1(u)$ and $\vartheta^2(u)f^2(u)$ appear in Fig. 3D. As it occurs for the membership functions and firing strength factors for this data the component products also intersect at value of $u_b = 3.98$. This estimate of $u_b$ is consistent with value previously reported by *Echavarría-Heras et al. (2019a)* for this data and deriving from conventional maximum likelihood biphasic regression. Figure 4A displays dispersion about resulting mean response function $v_{TSK}(u, \pi)$. Figure 4B shows residual scattering about the zero line. Region bounded by red lines determine (95%) confidence intervals. Figure 4C shows corresponding QQ plot.

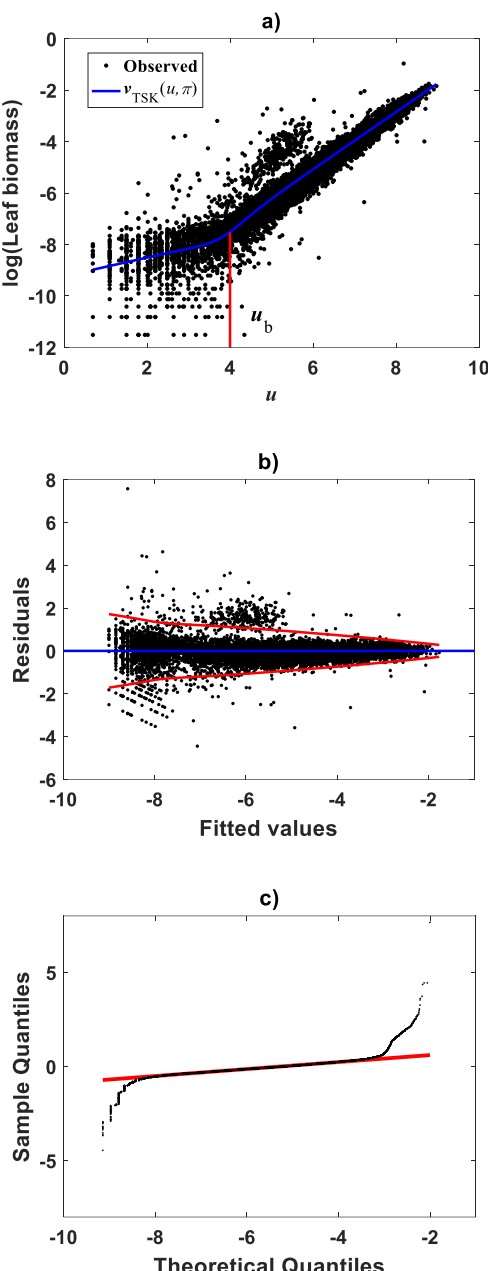

**Figure 4** **Fitting results of the TSK -PLA model for the** *Echavarría-Heras et al. (2019a)* **and** *Echavarría-Heras et al. (2018b)* **data set.** (A) shows the dispersion about the mean response curve identified through the regression model of Eq. (17) assuming heteroscedasticity in the form set by Eq. (35). (B) displays residual spread about the zero line. Region bounded by red lines determine (95%) confidence intervals. (C) presents corresponding QQ plot. Opposing a biased spreading about the mean response in Fig. 2A, distribution around the TSK-PLA mean response is fair all over the domain of the log transformed response. It is shown that the breaking point separates two phases conforming the identified non-log linear allometry.

**Table 4  Comparison of model performance metrics for TAMA and TSK-PLA models fitted on the *Echavarría-Heras et al. (2019a)* data set.** Included metrics are: AIC, CCC, $R^2$, SEE, MPE, and MPSE. Het refers to heteroscedastic and Hom to homoscedastic model.

| Method | $r_a$ | $q$ | AIC | $\rho_c$ | $R^2$ | SEE | MPE | MPSE |
|---|---|---|---|---|---|---|---|---|
| $v_s(u,\pi)$ | – | – | 17928.42 | 0.9375 | 0.8823 | 0.5723 | −0.2077 | 6.4777 |
| $v_{\mathrm{TSK}}(u,\pi)$ : Het | 0.47 | 2 | 16240.77 | 0.9475 | 0.9000 | 0.5276 | −0.1915 | 5.7908 |
| $v_{\mathrm{TSK}}(u,\pi)$ : Hom | 0.47 | 2 | 16237.40 | 0.9474 | 0.9000 | 0.5275 | −0.1915 | 5.8064 |

### Comparison TAMA vs. TSK–PLA

Compared with corresponding fitting results for the TAMA protocol (Fig. 2A) we can verify that plots in Fig. 4 show fairer residual spread patterns. Nevertheless, the QQ plot in Fig. 4C, still suggest deviation of $\epsilon_{TSK}$ residuals from a normal distribution pattern. Table 4 allows further comparison of performances of the TAMA and TSK proxies. This undoubtedly favor selection of the TSK scheme. Therefore, opposed to the linear model $v_s(u,\pi)$, the affine characterization of variability granted by $v_{TSK}(u,\pi)$ can better refer to inherent non-log linear allometry for the *Echavarría-Heras et al. (2019a)* data set. Certainly, the point $u_b = 3.98$ shown in Fig. 3B can be interpreted as a point separating two phases describing the variation pattern of the log transformed response $v$. One for small leaves valid over the region $u < u_c$ and another for large leaves over $u \geq u_b$. The form of the component products $\vartheta^i(u)f^i(u)$ shows that while $u$ drifts away from $u_b$ taking smaller and smaller values the closer the TSK output $v_{TSK}(u,\pi)$ will be to the component product $\vartheta^1(u)f^1(u)$. Conversely, the larger the distance between $u$ and $u_b$ for leaves in the large phase $u \geq u_b$ the closer $v_{TSK}(u,\pi)$ will be to $\vartheta^2(u)f^2(u)$. Relating to $E_s(v|u)$ shown in Fig. 2A, we can assess from Table 4 and Fig. 4A that the reproducibility strength of $E_{TSK}(v|u)$ is higher. Besides compared with Fig. 2B, the plot in Fig. 4B shows that distribution of residuals about the zero line for the TSK fit improved. Also compared to Fig. 2C, normal QQ plot in Fig. 4C shows a larger plateau where $\epsilon_{TSK}$ residuals track a normal distribution pattern. Nevertheless, application of an *Anderson & Darling (1952)* test to the residuals of regression Eq. (17) resulted in $AD = 370.17$. This associates a $p$-value $< 2.2 \times 10^{-16}$, that provides evidence against a normality assumption for the $\epsilon_{TSK}$ residuals. This is, in agreement with the observation that he normal Q–Q plot shown in Fig. 4C showing heavier tails than those expected for a normal distribution. It is worth pointing out that the break point $u_b$ identified by the fuzzy proxy $v_{TSK}(u,\pi)$ coincides with corresponding value obtained by *Echavarría-Heras et al. (2019a)* using conventional biphasic regression methods.

Correspondingly, Fig. 5A displays the plot of the estimated form of the mean response function $E_{gTSK}(x|y)$ of Eq. (19). Since, residuals $\epsilon_{TSK}$ are not normally distributed, Eq. (10) provided CF form. Figure 5B allows a visual assessment of the extent of biased projections in arithmetical scale tied to the TAMA surrogate $E_{gs}(x|y)$ calculated with Duan's form of $\delta$. Compared with spread deriving from the TSK model, TAMA's bias is significant. Besides, Table 4 allows assessment of differences in associated predictive strengths. All indices favor the TSK–PLA scheme. As suggested by perceptible bias shown in Fig. 5B, CCC value for TAMA's projections point to poor reproducibility of observed values. Besides, relevance

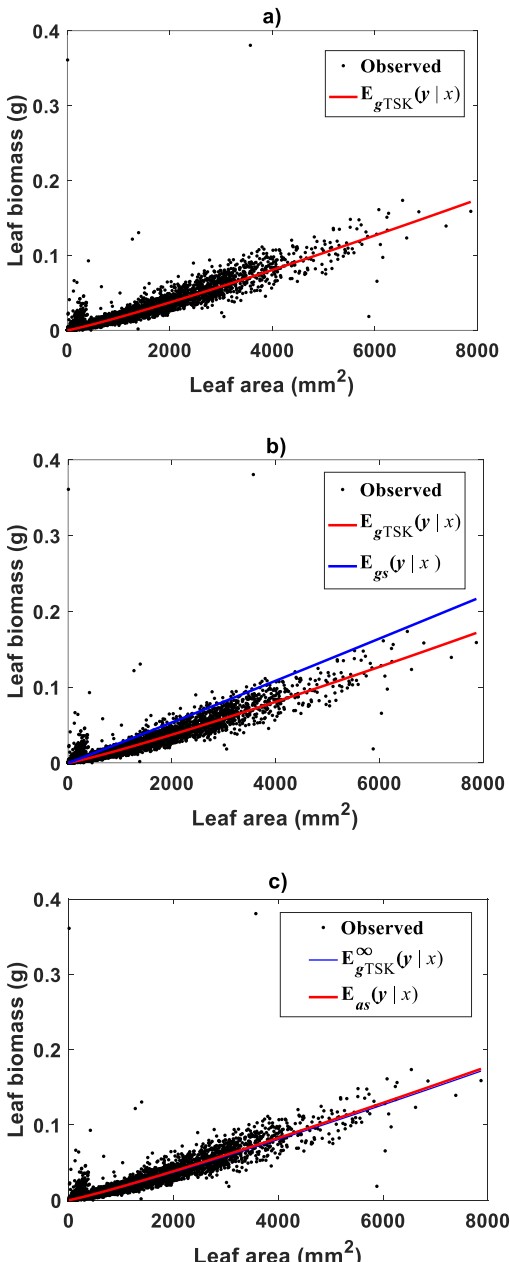

**Figure 5** **Comparison of TAMA and TSK-PLA mean responses in arithmetical scales fitted on the** *Echavarría-Heras et al. (2019a)* **data set.** (A) shows the distribution of observed eelgrass leaf biomass values $y$ about the mean response $E_{g_{TSK}}(y|x)$ (cf. Eq. (19)). Equation (10) provided the form of the correction factor. (B) shows the extent of bias tied to proxies $E_{gs}(y|x)$ calculated through the TAMA scheme and a Duan's form of the correction factor. (C) exhibits a remarkable correspondence between $E_{as}(y|x)$ derived from a fit of Huxley's formula of simple allometry and the asymptotic mean response derived from the TSK-PLA model (cf. Eq. (47)).

of accounting for curvature, this assessment highlights on the importance of choosing a proper form of $\delta$ for assuring consistency or retransformation results.
### Asymptotic analysis of TSK-PLA assembly

In this section we explain that adoption of a TSK-PLA approach allows exploration of asymptotic behavior of the allometric mean response function in arithmetical space. After replacing $f^1(u)$ and $f^2(u)$ as given by Eqs. (33) and (34) in Eq. (20) direct algebraic manipulation yields

$$v_{TSK}(u, \boldsymbol{\pi}) = ln\left(\beta_1^{(1-\vartheta^2(u))}\beta_2^{\vartheta^2(u)}\right) + (\alpha_1(1-\vartheta^2(u)) + \alpha_2\vartheta^2(u))u. \tag{36}$$

Similarly, it can be directly verified that firing strengths $\vartheta^1(u)$ and $\vartheta^2(u)$ as given by Eqs. (31) and (32) can be also expressed in the form

$$\vartheta^1(u) = \frac{1}{1 + e^{\tau(u,\theta,\lambda)}} \tag{37}$$

and

$$\vartheta^2(u) = \frac{e^{\tau(u,\theta,\lambda)}}{1 + e^{\tau(u,\theta,\lambda)}}, \tag{38}$$

where

$$\tau(u, \boldsymbol{\theta}, \boldsymbol{\lambda}) = \psi(\boldsymbol{\lambda})\phi(u, \boldsymbol{\theta}, \boldsymbol{\lambda}) + \xi(\boldsymbol{\theta}, \boldsymbol{\lambda}), \tag{39}$$

$$\psi(\boldsymbol{\lambda}) = \frac{(\lambda_2^2 - \lambda_1^2)}{2(\lambda_1\lambda_2)^2}, \tag{40}$$

$$\phi(u, \boldsymbol{\theta}, \boldsymbol{\lambda}) = [u + (\frac{\lambda_2^2\theta_1 - \lambda_1^2\theta_2}{\lambda_1^2 - \lambda_2^2})]^2 \tag{41}$$

$$\xi(\boldsymbol{\theta}, \boldsymbol{\lambda}) = \frac{(\lambda_2^2\theta_1 - \lambda_1^2\theta_2)^2 - (\lambda_1^2 - \lambda_2^2)[(\lambda_1\theta_2)^2 - (\lambda_2\theta_1)^2]}{2(\lambda_1^2 - \lambda_2^2)(\lambda_1\lambda_2)^2}. \tag{42}$$

with $\boldsymbol{\theta}, \boldsymbol{\lambda}$ standing for parameter vectors $(\theta_1, \theta_2)$ and $(\lambda_1, \lambda_2)$ one to one. We can then ascertain that $\phi(u, \boldsymbol{\theta}, \boldsymbol{\lambda})$ remains positive for all values of $u$. Also, the term $\xi(\boldsymbol{\theta}, \boldsymbol{\lambda})$, does not depend on $u$. Consequently, whenever the factor $\psi(\boldsymbol{\lambda})$ in Eq. (40) is positive, $\tau(u, \boldsymbol{\theta}, \boldsymbol{\lambda})$ will approach infinity as $u$ approaches infinity. Then, Eq. (37) implies $\vartheta^1(u)$ asymptotically vanishing as $u$ approaches infinity. Reversely, whenever $\psi(\boldsymbol{\lambda})$ is negative, the firing strength factor $\vartheta^1(u)$ will asymptotically approach one as $u$ approaches infinity. For the *Echavarría-Heras et al. (2019a)* data set we obtained $\psi(\boldsymbol{\lambda}) = 0.2756$, then we must have

$$\lim_{u \to \infty} \vartheta^1(u) = 0 \tag{43}$$

and since Eq. (A13) implies $\vartheta^2(u) = 1 - \vartheta^1(u)$, we also have

$$\lim_{u \to \infty} \vartheta^2(u) = 1. \tag{44}$$

Agreeing with Eqs. (19) and (20) back-transformation $e^v$ produces

$$E_{gTSK}\left(y|x\right) = \beta_1^{\left(1-\vartheta^2(u(x))\right)}\beta_2^{\vartheta^2(u(x))}a^{\left(\alpha_1\left(1-\vartheta^2(u(x))\right)+\alpha_2\vartheta^2(u(x))\right)}\delta. \tag{45}$$

We denote by means of the symbol, $E_{gTSK}^\infty\left(y|x\right)$ the limit of $E_{gTSK}\left(y|x\right)$ as $x$ approaches infinity, that is,

$$E_{gTSK}^\infty\left(y|x\right) = \lim_{x\to\infty} E_{gTSK}\left(y|x\right). \tag{46}$$

Then, Eqs. (31), (32) and (44) through (46) imply

$$E_{gTSK}^\infty(y|x) = \beta_2 x^{\alpha_2}\delta. \tag{47}$$

Then, the asymptotic mode $E_{gTSK}^\infty\left(y|x\right)$ identified for the *Echavarría-Heras et al. (2019a)* data set, is attains a form like Huxley's formula of simple allometry $w_s(x,\boldsymbol{p})$. Estimated parameters are $\alpha_2 = 1.1126$ and $\beta_2 = 7.8398 \times 10^{-6}$. Figure 5C displays observed leaf biomass values $y$ and their projections through the $E_{gTSK}^\infty\left(y|x\right)$ proxy. We can learn of a remarkable correspondence between the power function $w_s(x,\boldsymbol{p}) = \beta x^\alpha$ of Eq. (11) fitted by direct nonlinear regression methods and the asymptotic mean response $E_{gTSK}^\infty\left(y|x\right)$. Besides as established by Eq. (45) we can directly asses from Fig. 5C that for sufficiently large values of $x$ in the *Echavarría-Heras et al. (2019a)* data set, the mean response $E_{gTSK}\left(y|x\right)$ behaves as the power function $E_{gTSK}^\infty\left(y|x\right)$. Moreover, the order relationship $u \geq u_b$ holds for about 86% of analyzed data. This explains why corresponding phase of the TSK output can be considered dominant. This by the way elucidates the apparent benefit of fitting Huxley's formula of simple allometry by means of nonlinear regression model directly in arithmetical scale for the considered data. Indeed, such a fitting could deliver reasonable model adequacy results. But, as the present results show direct nonlinear examination based on Huxley's formula of simple allometry will fail to detect the different allometrical phases conforming the real variation pattern in the data. Then, as we have elaborated a log transformation step followed by nonlinear model identification in geometrical space could overcome the reproducibility deficiency of the TAMA approach.

**Fitting results of the TSK-PLA assembly: *Metrosideros polymorpha***
For trying the **main_fun_tsk_pla_model_fit.m** function on the *Mascaro et al. (2011)* data we set $r_a = 0.80$. This returned $q = 2$, heterogeneity. Figure 6 displays the plots of membership functions $\mu_{\Phi_i}(u)$, firing strength factors $\vartheta^i(u)$, consequent linear segments $f^i(u)$ and component products $\vartheta^i(u)f^i(u)$ identified by the fit of the TSK fuzzy model to the *Mascaro et al. (2011)* data set. Membership functions are shown in Fig. 6A. Fit suggests heterogeneity determined by a break point $u_b = 1.575$ as shown in Fig. 6B displaying firing strength factors. This estimate of $u_b$ is consistent with value previously reported by *Echavarría-Heras et al. (2019a)* for this data and deriving from conventional maximum likelihood biphasic regression. Break point suggest a growth phase $0 < u \leq u_b$ and a complementary $u > u_b$. We can interpreted these regions as dominance realms for the component product functions $\vartheta^1(u)f^1(u)$ and $\vartheta^2(u)f^2(u)$ one to one (Figs. 6C and 6D). Correspondingly, Fig. 7A shows spread about mean response function regions in geometrical space matching identified phases. Moreover, residual plot in Fig. 7B displays a

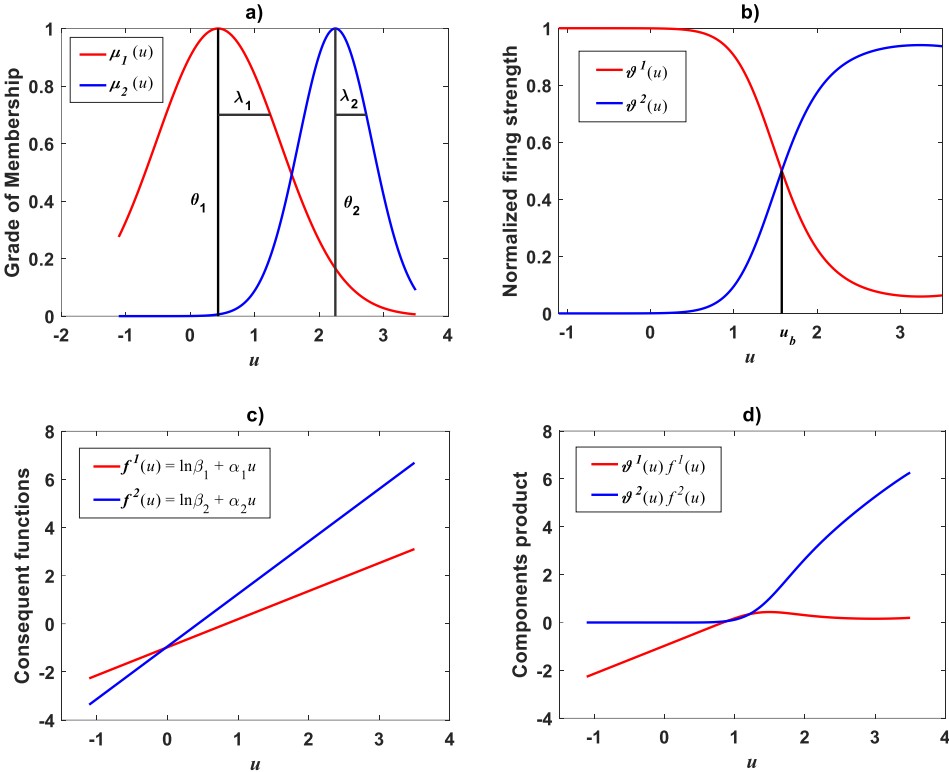

**Figure 6   Elements of the TSK-PLA fuzzy model identified on the *Mascaro et al. (2011)* data set.** Shown results associated to a value $r_a = 0.80$ for the clustering radius and corresponding to a $q = 2$, heterogeneity index. (A) plots of membership functions both given in the Gaussian form of Eq. (21). (B) presents plots of normalized firing strength factors given by Eqs. (31) and (32) one to one. A break point at $u_b = 1.575$ is shown. (C) displays consequent linear functions as given by Eqs. (33) and (34). (D) portraits component products.

fair distribution about the zero line. And, normal QQ-plot in Fig. 7C shows a large plateau where residuals track a normal distribution pattern. We can also ascertain from goodness of fit statistics in Table 5, that compared to the linear regression scheme of the TAMA protocol, the affine modeling approach composing the TSK-PLA scheme entails consistent identification of curvature in geometrical space.

## Identification of the TSK-PLA proxy: *Uca pugnax*

Huxley conceived a breakpoint in the log–log plot of chela mass vs. body mass of fiddler crabs (Uca pugnax) (*Huxley, 1924*; *Huxley, 1932*). Huxley situated this point between the 15th and 16th observations and assumed it meant a to a sudden change in relative growth of the chela approximately when crabs reach sexual maturity. Examination of Huxley's data by *Packard (2012a)* implied such a break point to be only putative and in Packard's own interpretation, perhaps explained by the fact that Huxley could have been misled by the effects of the log transformation itself, along with the format of graphical display that might have exaggerated the slopes of segments before and after the change point. In order to test the performance of the TSK-PLA protocol in analyzing Huxley's

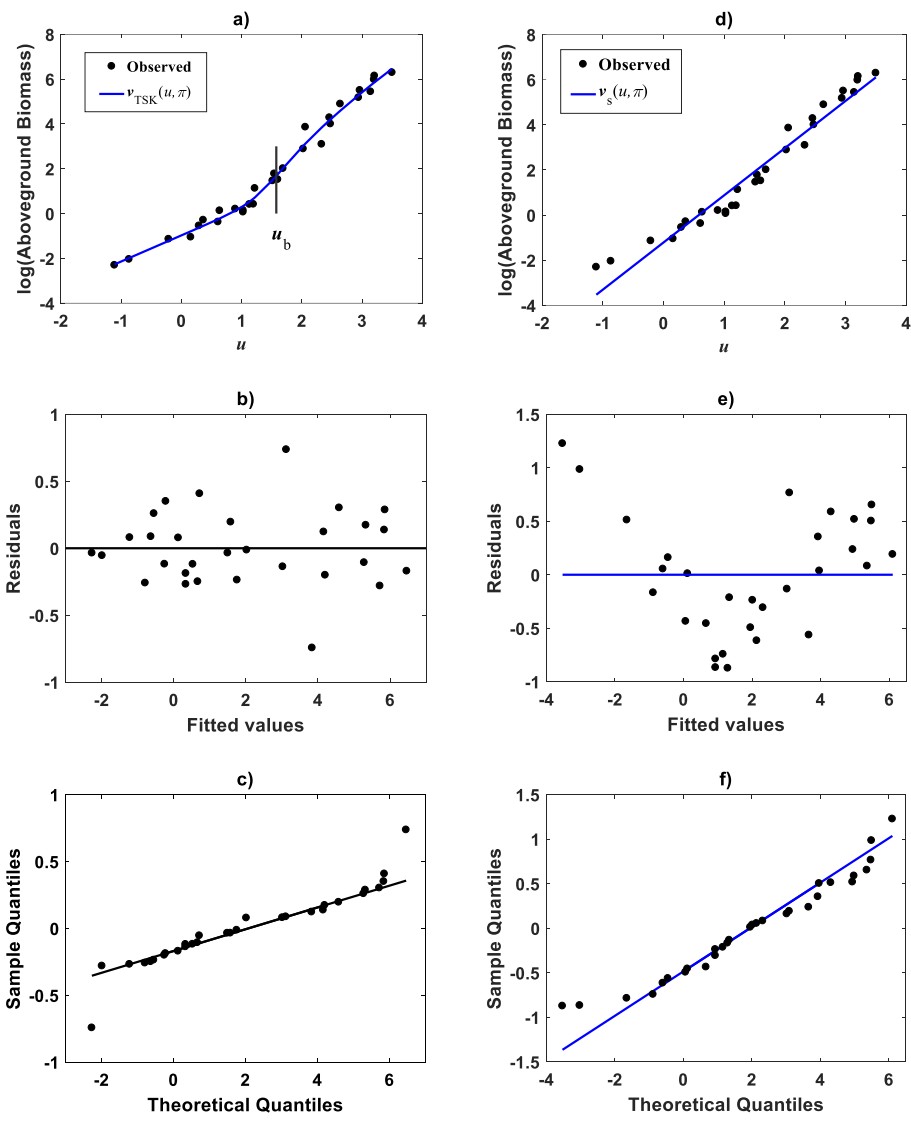

**Figure 7** **Spread plots for the TSK-PLA model fitted on the *Mascaro et al. (2011)* data set.** The spread about the TSK-PLA mean response displays remarkable reproducibility and consistency of biphasic allometry (A). The residual plot displays a fair spread about the zero line (B). The Normal-QQ plot shows a large plateau where residuals track a normal distribution pattern (C). (D–F) show the spread about mean response, residual and QQ-plot of TAMA's fit to this data one to one.

*Uca pugnax* data, we took averages of both body mass and chela mass form Table 1 in Huxley's report (*Huxley, 1932*). Concurrent log transformed values appear in Fig. 1C. For easy of presentation a break point as conceived by Huxley's will be denoted here through the symbol $u_{bH}$. One substantial advantage of the fuzzy logic approach over conventional probabilistic slants is that the former facilitates knowledge based modeling. In order to incorporate previous knowledge, we we abided by Huxley's assertion of biphasic allometry in *Uca pugnax*. Then, we examined heterogeneity patterns predicted by the TSK-PLA system for different values of clustering radius $r_a$. Particularly, setting $r_a = 0.8$

**Table 5  Model performance metrics for TAMA and TSK-PLA models fitted on the *Mascaro et al. (2011)* data set.** Included metrics are: AIC, CCC, $R^2$, SEE, MPE, and MPSE.

| Method | $r_a$ | $q$ | AIC | $\rho_c$ | $R^2$ | SEE | MPE | MPSE |
|---|---|---|---|---|---|---|---|---|
| $v_s(u,\pi)$ | —- | — | 53.4608 | 0.9767 | 0.9544 | 0.5712 | 10.5825 | 47.7494 |
| $v_{\mathrm{TSK}}(u,\pi)$ | 0.80 | 2 | 23.4700 | 0.9943 | 0.9888 | 0.3200 | 5.9292 | 24.4923 |

returned $q = 2$, arranging biphasic allometry. Acquired firing strengths appear in Fig. 8A, exhibiting a break point at $u_b = 5.831$. Figure 8B display consequent linear functions with estimated slopes $\alpha_1 = 1.2676$ and $\alpha_2 = 1.4708$ one to one respectively. In the settings of performed TSK-PLA analysis these correspond to exponents characterizing allometric phases as conceived in Huxley's original theoretical standpoint. Correspondingly, Fig. 8C portrays consequent component products $\vartheta^1(u)f^1(u)$ and $\vartheta^2(u)f^2(u)$. Similarly, Fig. 8D shows spread about mean response $v_{TSK}(u,\boldsymbol{\pi})$ including placement of $u_b$ in a display in compliance with that in Fig. 3 in *Huxley (1932)*. We can be aware that location of $u_b$ is shifted back relative to $u_{bH}$. Figure 8E displays placement of $u_b$ and spread about $v_{TSK}(u,\boldsymbol{\pi})$ in the original scale of data (cf. Fig. 1C). But instead, we may integrate previous knowledge by considering for that the break point $u_{bH}$ actually exists. Then, we can search among different values of $r_a$, the one for what the TSK-PLA arrangement compromises a break point $u_b$ placed as $u_{bH}$ and also a maximum reproducibility strength of interpolation by $v_{TSK}(u,\pi)$. Accordingly, setting $r_a = 0.2$ brought about $q = 7$ sub models, inducing a maximum reproducibility strength of interpolation function $v_{TSK}(u,\pi)$ and where $u_{bI}$, one of six detected break points is placing as $u_{bH}$. (Fig. 8E and Table 6). Interestingly, visual examination of plot showing $v_{TSK}(u,\pi)$ suggests a pattern accommodating two linear segments that alternate about $u_{bI}$. Moreover, using the interpolation points $(u, v_{TSK}(u,\pi))$ we fit two linear segments of slopes $\alpha_{1I} = 1.626$ and $\alpha_{2I} = 1.274$ before and after $u_{bI}$ one to one (Fig. 8F). Since $u_{bI}$. can be taken as a proxy for $u_{bH}$ the TSK-PLA interpolation mode could suggest Huxley's reasoning of biphasic allometry in in *Uca pugnax* as consistent. In the meantime acquired $q = 7$, interpolation confirms the outstanding capabilities of the TSK-PLA device to approximate the dynamics of the logtransformmed allometric response. This can be better ascertained from Fig. 9A through Fig. 9C presenting spread about the high order interpolation function $v_{TSK}(u,\pi)$, as well as, concomitant residual and QQ plots in conforming order. Moreover, Fig. 9D through Fig. 9F allow visual comparison of parallel results by a TAMA's fit. Additionally, Table 6 compares model performance metrics for the TSK-PLA interpolation and TAMA's output fits. We can ascertain that the TSK-PLA interpolation stands a better fit. In any event, the non-probabilistic interpretation of uncertainty backing the TSK–PLA approach seems to advocate biphasic heterogeneity in geometrical space for Huxley's *Uca pugnax* data.

## Fitting results of the TSK- PLA assembly: *Gadus chalcogrammu*

A fit of the TSK-PLA protocol to *Gadus chalcogrammu* data reported in *De Robertis & Williams (2008)*, can exhibit reliability of this paradigm in further way. Visual examination of spread in geometrical space may suggest curvature. But, setting $r_a = 0.5$ led to highest reproducibility of $v_{TSK}(u,\pi)$ characterized in a linear form. Indeed, plots in Fig. 10 show

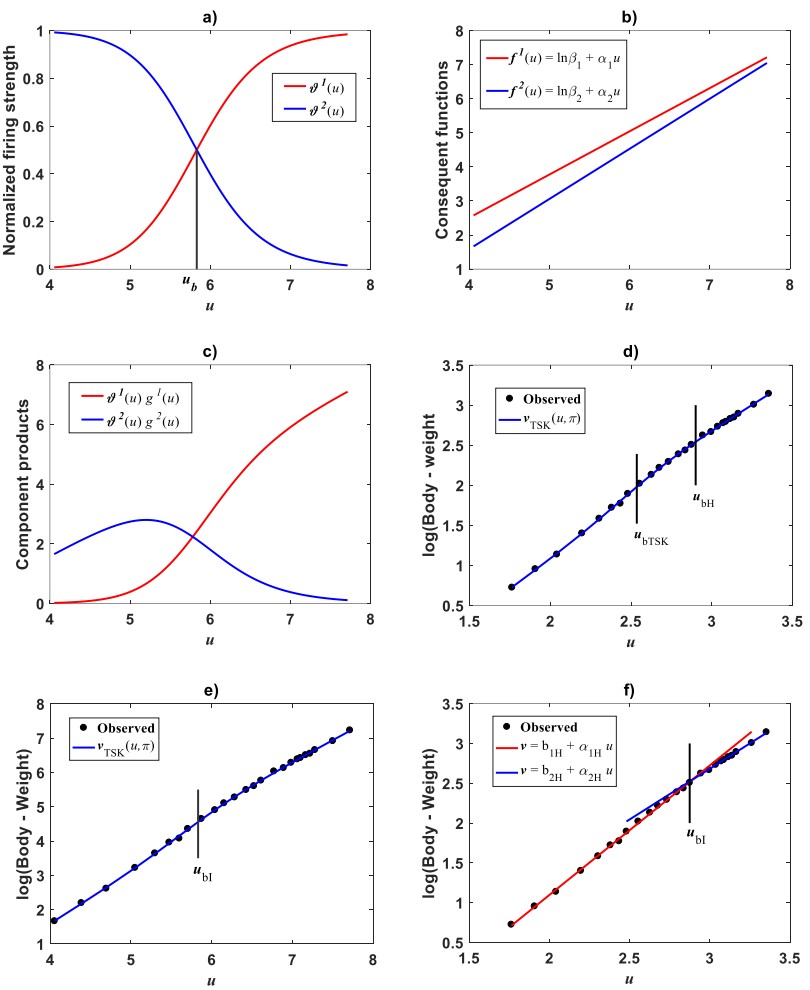

**Figure 8** **TSK-PLA model identified on *Huxley (1932)* *Uca pugnax* data.** For $r_a = 0.8$ the fuzzy inference system returned $q = 2$ heterogeneity. (A) exhibits firing strengths intersecting at a break point $u_b = 5.813$ in original log scales. (B) acquired linear consequents. (C) component products. (D) shows position of $u_b$ relative to Huxley's break point $u_{bH}$ in a display conforming that in Fig. 3 of *Huxley (1932)*. (E) spread about TSK-PLA interpolation function produced by $r_a = 0.2$ and $q = 2$ in original log scales. This plot shows $u_{bI} = 6.78$ one of detected breakpoints. This can be considered as a proxy for Huxley's designated break point $u_{bH}$. Interpolation results in (E) suggest the biphasic arrangement of linear segments about $u_{bI}$ shown in (F).

**Table 6** **Model performance metrics for TAMA and TSK-PLA models fitted on the *Huxley (1932)* *Uca pugnax* data set.** Included metrics are: AIC, CCC, $R^2$, SEE, MPE, and MPSE.

| Method | $r_a$ | $q$ | AIC | $\rho_c$ | $R^2$ | SEE | MPE | MPSE |
|---|---|---|---|---|---|---|---|---|
| $v_s(u,\pi)$ | – | – | $-51.4477$ | 0.9986 | 0.9972 | 0.0832 | 0.6519 | 1.5399 |
| $v_{\mathrm{TSK}}(u,\pi)$ | 0.8 | 2 | $-97.8184$ | 0.9999 | 0.9997 | 0.0301 | 0.2359 | 0.4239 |
| $v_{\mathrm{TSK}}(u,\pi)$ | 0.2 | 7 | $-127.57$ | 0.9999 | 0.9999 | 0.0166 | 0.1301 | 0.2058 |

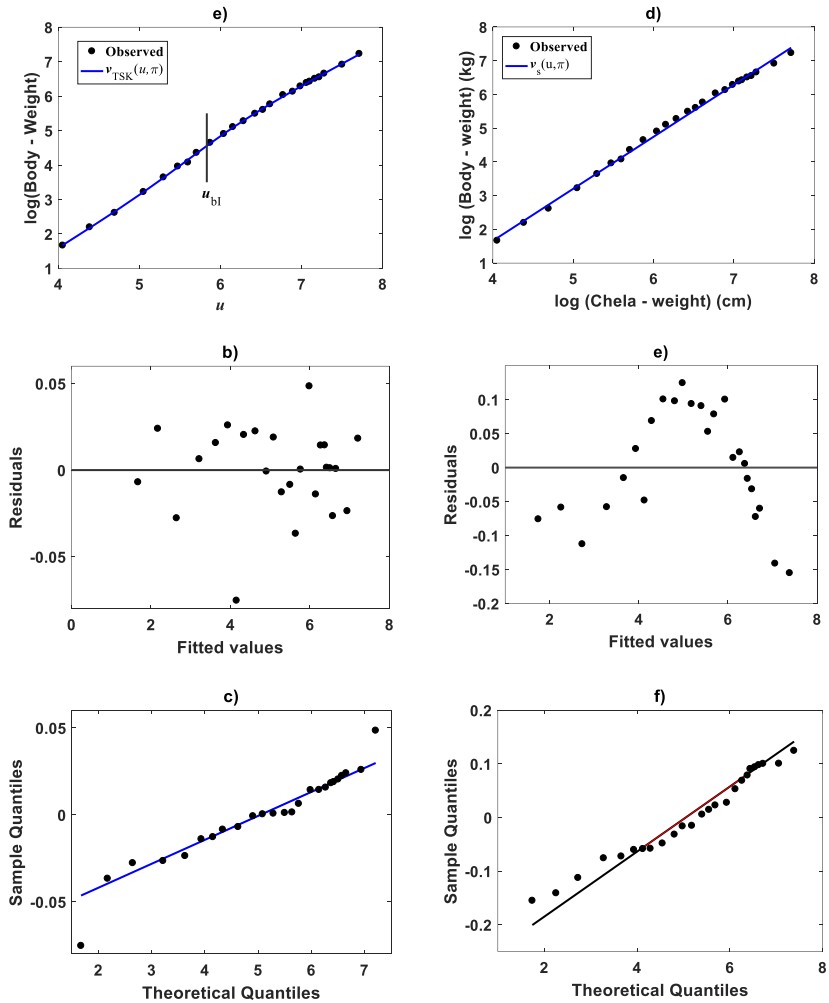

**Figure 9** **Comparison TAMA vs. TSK-PLA fuzzy model fitted on** *Huxley (1932)* *Uca pugnax* **data.** (A) exhibits spread about TSK-PLA mean response as determined by a $r_a = 0.8$ and $q = 2$ fit on Huxley's *Uca pugnax* data. Associating residual and QQ-plots are shown in (B) and (C) one to one. (D) trough (F) display corresponding spreads produced by TAMA's fit to referred data.

that identification of the TSK-PLA model for this data, produced only one membership function $\mu_{\Phi_1}(u)$ (Fig. 10A). This corresponds to a firing strength $\vartheta^1(u)$ set to one (Fig. 10B), and a conforming single TAMA's form linear consequent $f^1(u)$ (Fig. 10C). This matched the single linear component product function shown in Fig. 10D. As a result, no heterogeneity as determined by breaking points $u_b$ was detected for this data. Consequently, the TSK arrangement suggests a fit equivalent to the TAMA approach. Moreover, spread abut mean response, residual and Normal QQ-plots for a TAMA fit performed in this data (Fig. 11A through Fig. 11C respectively) seem to faithfully agree to corresponding plots (Fig. 11D, through Fig. 11F) associating to the TSK-PLA fit. In turn model performance metrics in Table 7 corroborate these alternate fits as equivalent. Therefore, the TSK-PLA

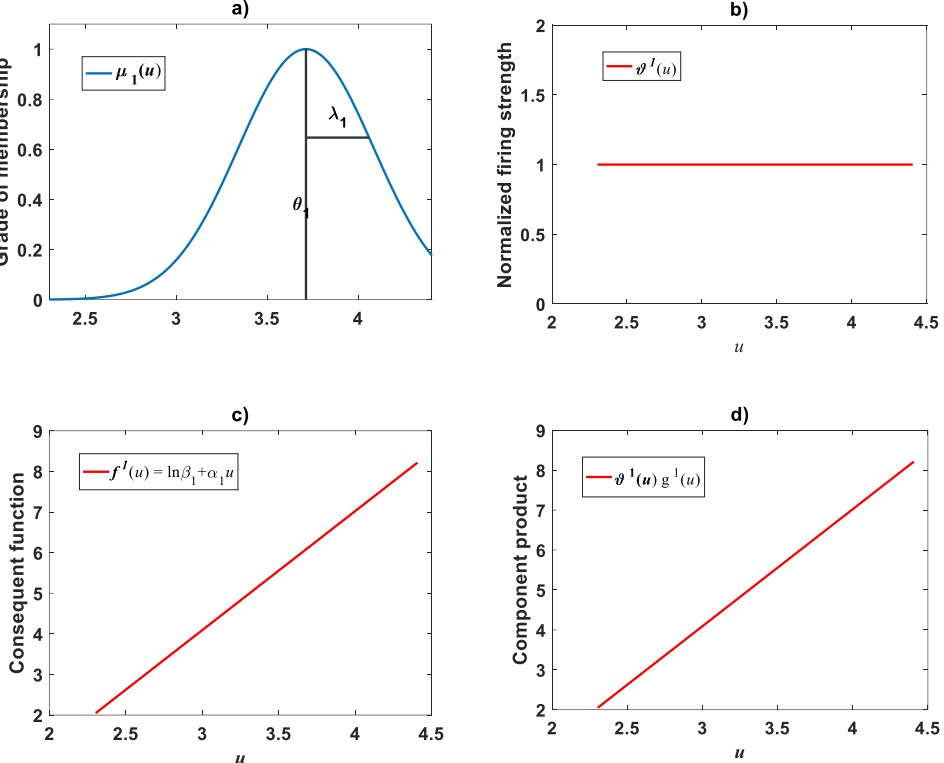

**Figure 10** Elements of the TSK-PLA model identified on the *De Robertis & Williams (2008)* **data.** A single membership function in the Gaussian form given by Eq. (21) is shown in (A). Corresponding firing strength is displayed in (B). Consequent function appears in (C). Component product appears in (D). These components rule out nonlinearity in geometrical space, suggesting consistency of a TAMA approach.

assembly seemingly adapts required complexity. This supports judgement on this paradigm being considered as a generalized tool for allometric examination in geometrical space.

## Assembly of the TSK- MPCA proxy

We assume that $w(x, \boldsymbol{p})$ as intended for MPCA can be modeled by $w_{TSK}(x, \boldsymbol{p})$ as expressed by means of Eq. (A14), in arithmetical space, namely

$$w_{TSK}(x, \boldsymbol{p}) = \sum_{1}^{q} \vartheta^i(x) f^i(x) \tag{48}$$

with firing strengths $\vartheta^i(x)$ given by

$$\vartheta^i(x) = \frac{\mu_{\Phi_i}(x)}{\sum_{1}^{q} \mu_{\Phi_k}(x)} \tag{49}$$

being $\mu_{\Phi_i}(x)$ for $i = 1, 2 \ldots \ldots q$ the involved membership functions. We also undertake that both $w(x, \boldsymbol{p})$ and $x$ remain positive, and that

$$\lim_{x \to 0^+} w(x, \boldsymbol{p}) = 0. \tag{50}$$

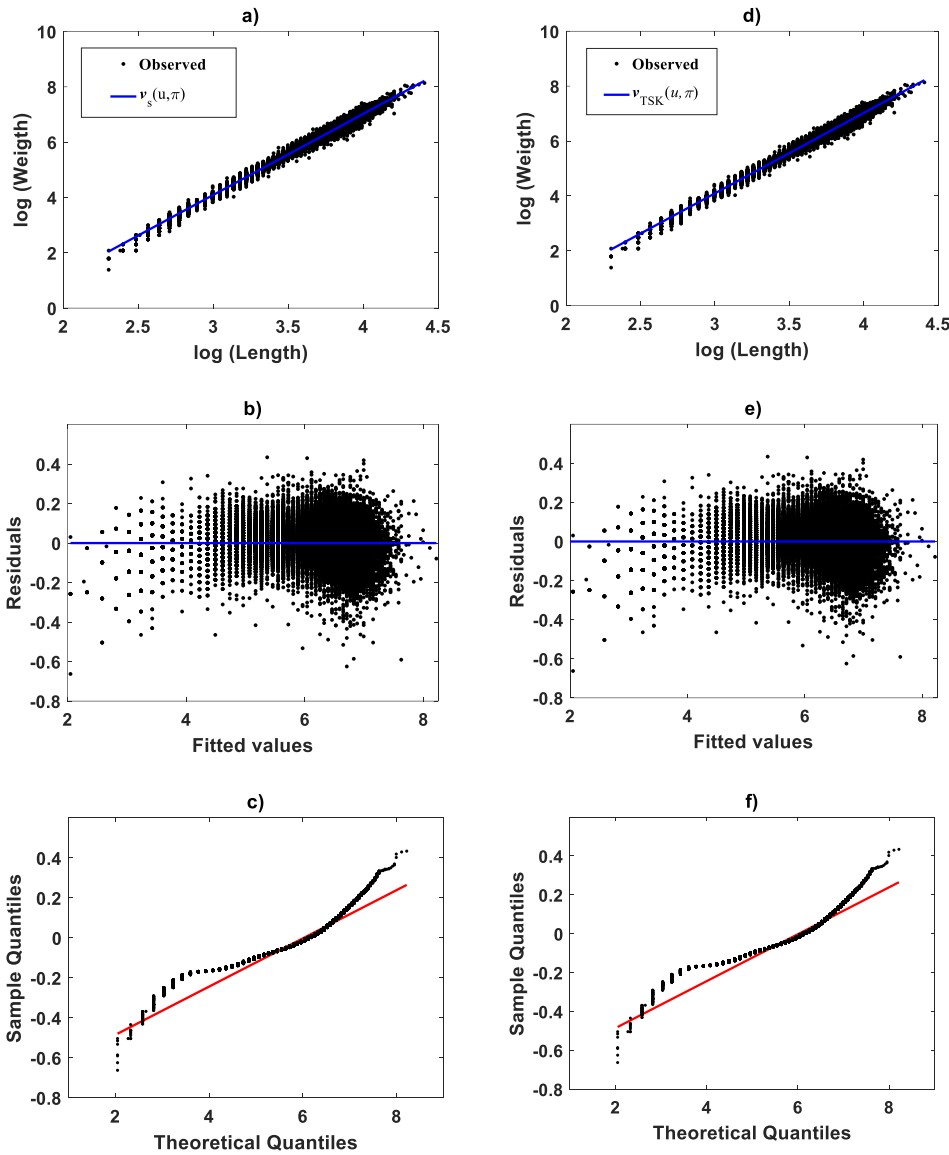

**Figure 11  Comparison of TAMA and TSK-PLA fuzzy model fitted on the *De Robertis & Williams (2008)* data set.** (A–C) display spread about mean response, residual plot and QQ-Normal plot for the fit of the TAMA protocol one to one. (D–F) present corresponding plots for the fit of the TSK fuzzy model.

**Table 7  Model performance metrics for TAMA and TSK-PLA models fitted on the *De Robertis & Williams (2008)* data set.** Included metrics are: AIC, CCC, $R^2$, SEE, MPE, and MPSE.

| Method | $r_a$ | $q$ | AIC | $\rho_c$ | $R^2$ | SEE | MPE | MPSE |
|---|---|---|---|---|---|---|---|---|
| $v_s(u,\pi)$ | – | – | −55181 | 0.9940 | 0.9881 | 0.0945 | 0.0185 | 1.274 |
| $v_{\text{TSK}}(u,\pi)$ | 0.5 | 1 | -55169 | 0.9941 | 0.9882 | 0.0946 | 0.0186 | 1.274 |

This sets the imput space $X$ to be $R^+$. We then contemplate that membership functions can be expressed through a composite log normal form that satisfies the constrain by Eq. (50), namely

$$\mu_{\Phi_i}(x) = c(e^{h_i(x)} - 1) \tag{51}$$

where $c = (1 - e)^{-1}$ and

$$h_i(a) = e^{\left\{-\frac{1}{2}\left[\left(\frac{\ln a - \theta_i}{\lambda_i}\right)^2\right]\right\}} \tag{52}$$

with $\theta_i$ and $\lambda_i$ for $i = 1, 2, \ldots., q$ parameters. Correspondingly we consider the consequents $f^i(x)$ to be linear functions, that is,

$$f^i(x) = c_{i1} + c_{i2}x. \tag{53}$$

It is worth recalling that Eq. (15) provides the form of linked regression protocol.

Identification of $w_{TSK}(x, p)$ as given by Eq. (48) through Eq. (53) from data pairs $(y, x)$ in direct scale is performed by means of the Matlab function **main_fun_tsk_mpca_model_fit.m** supplied in the supplemental files section. Heterogeneity and reproducibility strength features of $w_{TSK}(x, p)$ can be explored in an interactive way through different characterizations of the clustering radius parameter $r_a$ as specified by Eq. (B7) through Eq. (B9).

## Identification of the TSK–MPCA proxy: *Zostera marina*

For the *Echavarría-Heras et al. (2019a)* data a try of the **main_fun_tsk_mpca_model_fit.m** code setting $r_a = 0.5416$ returned $q = 2$ for a biphasic mode and a maximum reproducibility of $w_{TSK}(x, p)$. Figure 12A displays acquired firing strength functions. The estimated break point was $x_b = 49.632$ consistent with the retransformed value of $u_b = 3.9$ for this data set. This means that variability of the response $y$ indeed conforms to a MPCA pattern in the direct scale of data. Corresponding spread about fitted mean response function $E_{aTSK}(y|x)$ appear in Fig. 12B. This plot allows comparison to its counterpart $E_{gTSK}(y|x)$ produced by retransformation of mean function $v_{TSK}(u, \pi)$ to arithmetical space. We can be aware of remarkable correspondence through $x$ values. This validates adequacy of a TSK-PLA analysis for this data. Figures 12C and 12D show residual spread and QQ-plot for the TSK-MPCA fit. Figures 12E and 12F show corresponding plots for retransformed TSK-PLA fit. Besides Table 8 allows assessment of addressed proxies through model performance metrics. This could endure a judgement that concurrent MPCA pattern in arithmetical space can be consistently characterized by retransformation of PLA results.

## Identification of the TSK-MPCA proxy: *Metrosideros polymorpha*

Correspondingly, taking as previous knowledge a manifestation of biphasic allometry as detected by the TSK-PLA scheme, for the *Mascaro et al. (2011)*, we examined the possibility of the TSK-MPCA arrangement identifying a similar pattern in direct scales. Indeed, by setting $r_a = 0.855$ the **main_fun_tsk_mpca_model_fit.m** function returned $q = 2$ for a biphasic mode and a $w_{TSK}(x, p)$ of reliable reproducibility. Identified firing strengths, display in Fig. 13A. Again analysis in direct scale detected by the TSK-MPCA approach

Peer

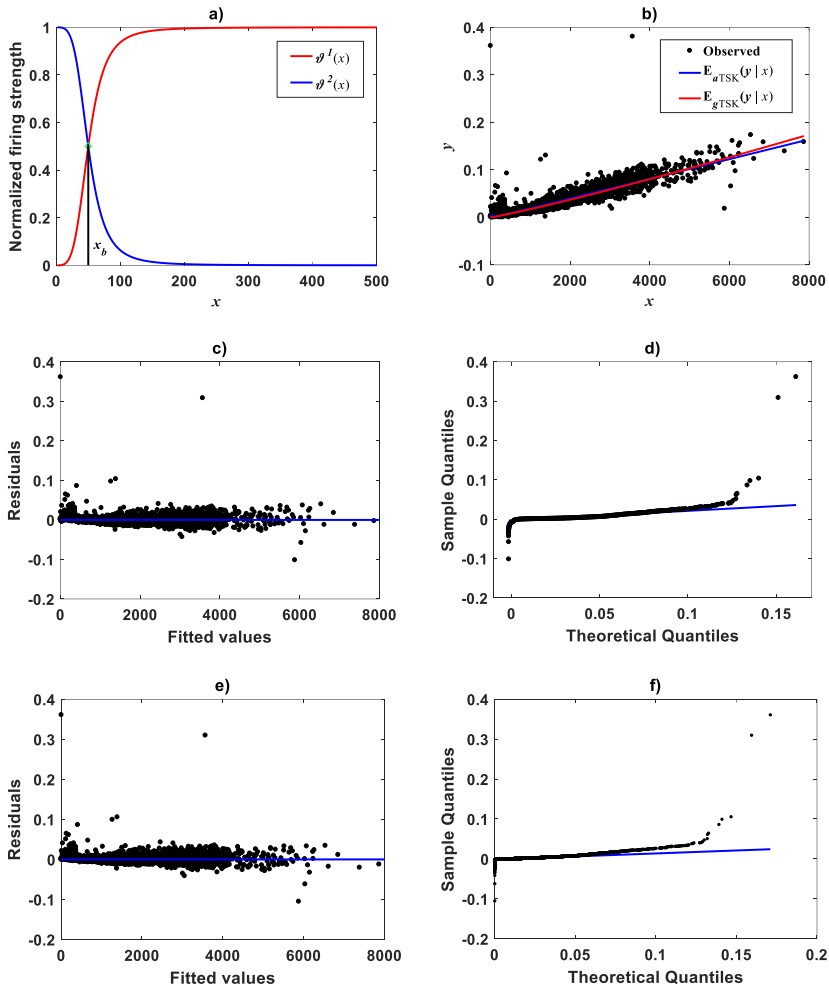

**Figure 12    TSK-MPCA fitted on the *Echavarría-Heras et al. (2019a)*.** Setting $r_a = 0.5416$ returned $q = 2$ for present eelgrass data analyzed by means of the TSK-MPCA fuzzy model of Eq. (48) through Eq. (53). (A) firing strength factors detecting a break point placed at $x_b = 49.632$. (B) spread about fitted mean response function $E_{aTSK}(y|x)$ compared to $E_{gTSK}(y|x)$ derived from retransformation of the TSK-PLA output. We can be aware that reproducibility strengths are equivalent. This can be stressed by performance metrics in Table 8. (C) through (D) presenting residual and QQ-plots confirm equivalence of $E_{aTSK}(y|x)$ and $E_{gTSK}(y|x)$.

corroborates the consistency of break point allometry assumption for this data. We can learn of a break point estimated at $a_b = 8.8662$. This estimate is consistent with resulting from a two linear segment mixture regression model performed by present authors. Spreads about fitted mean functions shown in Fig. 13B reveal remarkable correspondence of projections by $E_{aTSK}(y|x)$ and $E_{gTSK}(y|x)$. This can be stressed by performance metrics in Table 9. In turn this demonstrates adequacy of a TSK-PLA approach for the analysis of this data. Figures 13C and 13D display residual and QQ plots for TSK-MPCA fit to *Metrosideros polymorpha*. Equivalent plots associating to the retransformed TSK- PLA are displayed in Figs. 13E and 13F, correspondingly. Exploring interpolation capabilities of the TSK-MPCA

**Table 8  Model performance metrics for TSK-PLA and TSK-MPCA models fitted on the *Echavarria-Heras et al. (2019a)* data set.** Included metrics are: AIC, CCC, $R^2$, SEE, MPE, and MPSE.

| Method | $r_a$ | $q$ | AIC | $\rho_c$ | $R^2$ | SEE | MPE | MPSE |
|---|---|---|---|---|---|---|---|---|
| $E_{aTSK}(y\|x)$ | 0.5416 | 2 | −73038.16 | 0.9294 | 0.8678 | 0.007 | 1.0993 | 59.31 |
| $E_{gTSK}(y\|x)$ | 0.47 | 2 | −73107.75 | 0.9282 | 0.8688 | 0.007 | 1.0956 | 63.00 |

for this data led to considering an alternate clustering radius set at a value $r_a = 0.52$. Resulting heterogeneity index was $q = 3$ that resulted in good model assessment metrics (Table 9) and a break point placed at $x_b = 4.832$. This is in agreement with retransformed TSK-PLA estimation for this data. Nevertheless, forcing an interpolation mode of the TSK-MPCA to achieve a break point placed in agreement with previous estimation brings about complexity that renders biological interpretation difficult.

### Identification of the TSK-MPCA proxy: *Uca pugnax*

Firing strengths, of a $r_a = 0.668$, $q = 2$, TSK-MPCA fit to Huxley's *Uca pugnax* data set are displayed in Fig. 14A. We can be aware of heterogeneity as corresponds to dominance of sub models before and after the break point placed at $x_b = 340.7$, matching $exp(u_b)$ with $u_b = 5.83$ the break point determined by a TSK-PLA fit to this data. Figure 14B show spread about resulting mean function $E_{aTSK}(y|x)$ and compares to $E_{gTSK}(y|x)$ gotten by retransformation of fitted TSK-PLA. Plot suggest equivalent reproducibility strengths. Nevertheless as it can be made certain by model performance metrics in Table 10 the $E_{gTSK}(y|x)$ proxy entails relatively higher reliability. Figures 14C and 14D one to one show residual and QQ plots corresponding to the TSK-MPCA fit. Similarly, residual and QQ plots for the back-transformed TSK-PLA fit appear in Figs. 14E and 14F. Table 10 also includes performance metrics for $E_{gTSK}(y|x)$ gotten by retransforming the output of the $r_a = 0.2$ and $q = 7$, fit of the TSK-PLA. We can assert that resulting interpolation $E_{gTSK}(y|x)$ yields a relatively better fit. Therefore, results of the retransformed form of a TSK-PLA approach entails consistent results in direct scales. In other words, logtransformation based procedures do not lead to biased results for this data. But above all, results of a TSK-MPCA fit could provide a clue clearing an apparent misinterpretation of Huxley about existence of a break point in his analysis of *Uca pugnax* data. Indeed, as stated above we have $u_b = \ln(a_b)$. This implies $u_b$ being the image of $a_b$ under logtransformation. Then, claiming existence of $u_b$ attributable to distortion set by a logtransformation itself is inappropriate. Agreeing with *Packard (2012a)*, we have no doubt that conventional statistical methods do not put up with existence of $u_b$ as detected by the present fuzzy inference system. But, this fact cannot be exhibited to question fuzzy methods. These relying in non-probabilistic approaches have provided reliable interpretation of uncertainty as it can be inferred by fuzzy approach solutions to many problems of identification and control of nonlinear systems.

### Identification of the TSK-MPCA proxy: *Gadus chalcogrammu*

When we assessed the performance of the TSK-MPCA device on the *De Robertis & Williams (2008)* data we found results consistent to the TSK-PLA fit reported above. Indeed, a TSK-MPCA analysis based on $r_a = 0.50$ for this data returned $q = 1$, for a single membership

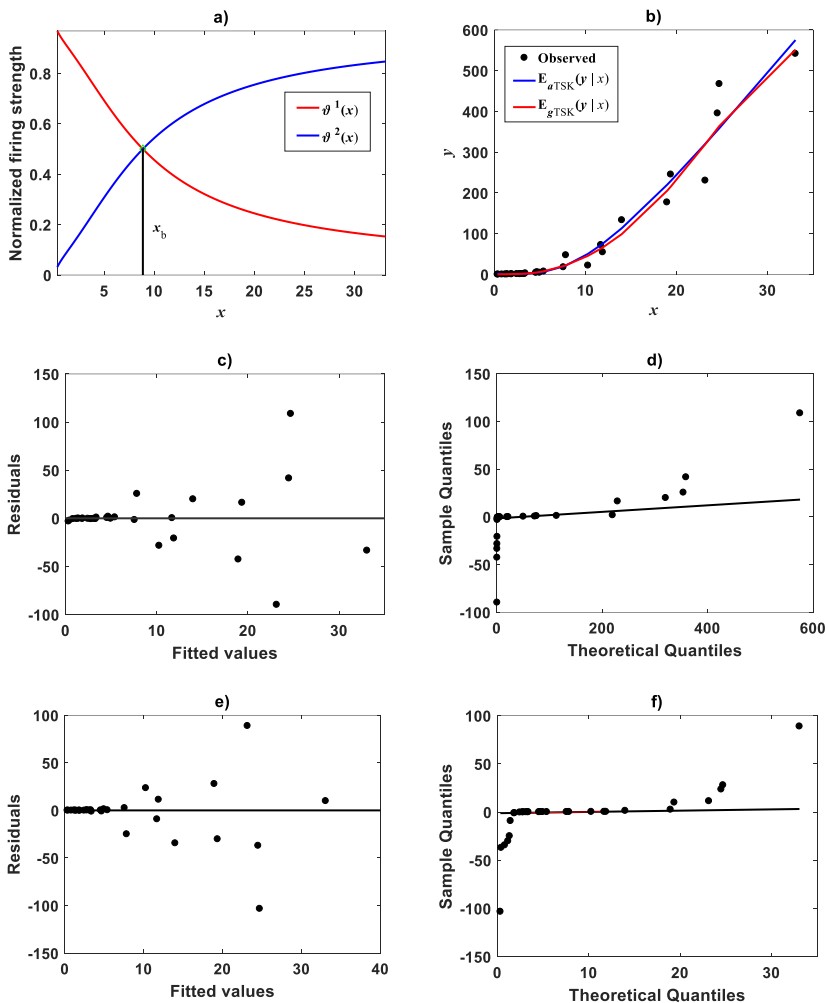

**Figure 13** **TSK-MPCA fitted on *Mascaro et al. (2011)* data set.** (A) displays the $q = 2$ firing strength factors deriving from a $r_a = 0.855$ of the TSK-MPCA fit. We can learn of a break point estimated at $x_b = 8.8662$. Spreads about fitted mean functions shown in Fig. 13B reveal remarkable correspondence of projections by $E_{aTSK}(y|x)$ and $E_{gTSK}(y|x)$. This can be stressed by performance metrics in Table 9. (C) and (D) residual and QQ plots for TSK-MPCA fit one to one. Equivalent plots for retransformed results of TSK-PLA fitted on this data are displayed in (E) and (F).

function. This consequently associates to a single firing strength $\vartheta^1(x) = 1$. As a result, we have to contemplate a single component product of a linear form in Eq. (48). No break point composed heterogeneity in direct scales can be verified for this data. Moreover, implied linear form of Eq. (48) does not fit required complexity in direct scales. Nevertheless, previous knowledge on consistency of corresponding TSK-PLA fit suggest using the interpolation features of TSK-MPCA to grant adequacy. For this empirical aim, for instance taking the clustering parameter $r_a = 0.22$ in Eq. (B7) we can manage to obtain an heterogeneity index of $q = 3$. This entails three sub models composing Eq. (48). Figures 15A through 15C display spread about resulting form of interpolation mean response function

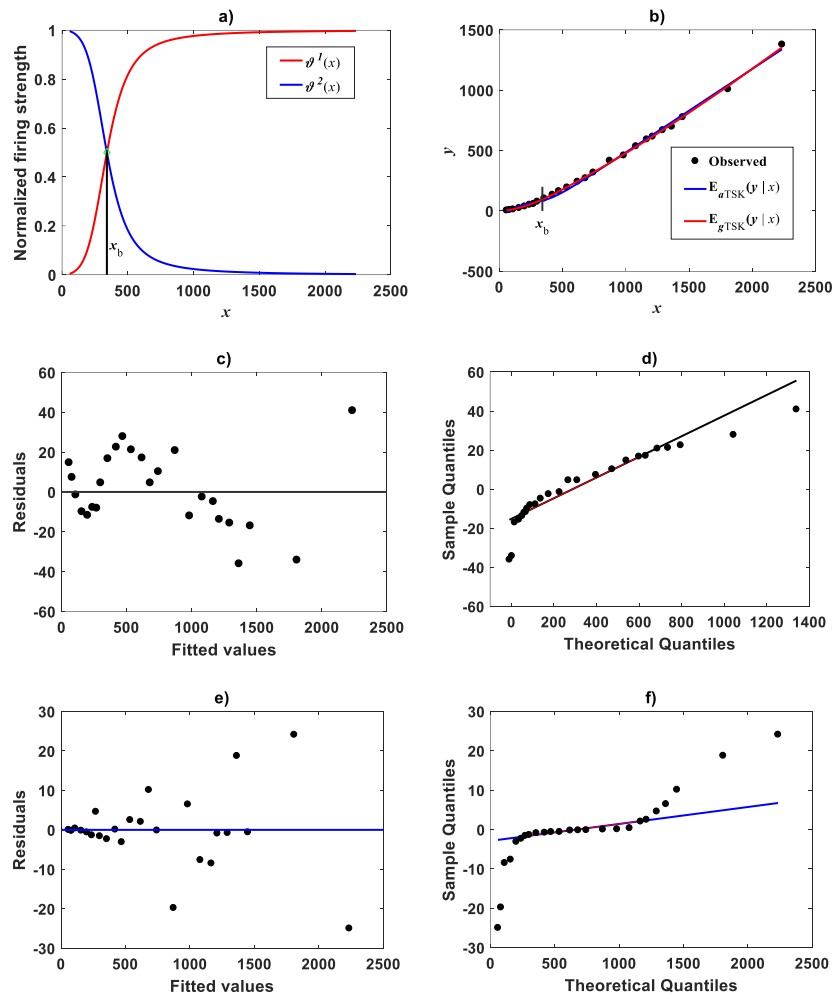

**Figure 14    TSK-MPCA fitted on *Huxley (1932) Uca pugnax* data.** (A) displays the $q = 2$ firing strength factors deriving from an $r_a = 0.668$ fit of the TSK-MPCA. A break point places at $x_b = 340.7$. (B) shows spread about resulting mean function $E_{aTSK}(y|x)$ and compares to $E_{gTSK}(y|x)$ gotten by retransformation of fitted TSK-PLA. Plot suggest corresponding reproducibility strengths. Nevertheless, as it can be made certain by model performance metrics in Table 10, the $E_{gTSK}(y|x)$ proxy entails relatively higher reliability. (C) and (D) display residual and QQ plots one to one for $E_{aTSK}(y|x)$. (E) and (F) show corresponding plots for $E_{gTSK}(y|x)$. We may be aware that log-transformation based procedures do not lead to biased results for this data.

$E_{aTSK}(y|x)$, as well as, residual and normal QQ plots in that order. Figure 15A also allows visual appraisal of a better reproducibility by $E_{gTSK}(y|x)$. Figure 15D presents spread about $E_{as}(y|x)$ and compares to $E_{gTSK}(y|x)$. We can observe that both proxies entitle similar reproducibilities. Figures 15E and 15F present residual spread and QQ plot accompanying $E_{as}(y|x)$. Table 11 compares reproducibility metrics for the $E_{aTSK}(y|x)$, $E_{gTSK}(y|x)$ and $E_{as}(y|x)$ proxies for this data. Again confrontation of model performance metrics shows that retransformation of the TSK- PLA output stands reliable results in direct scales.

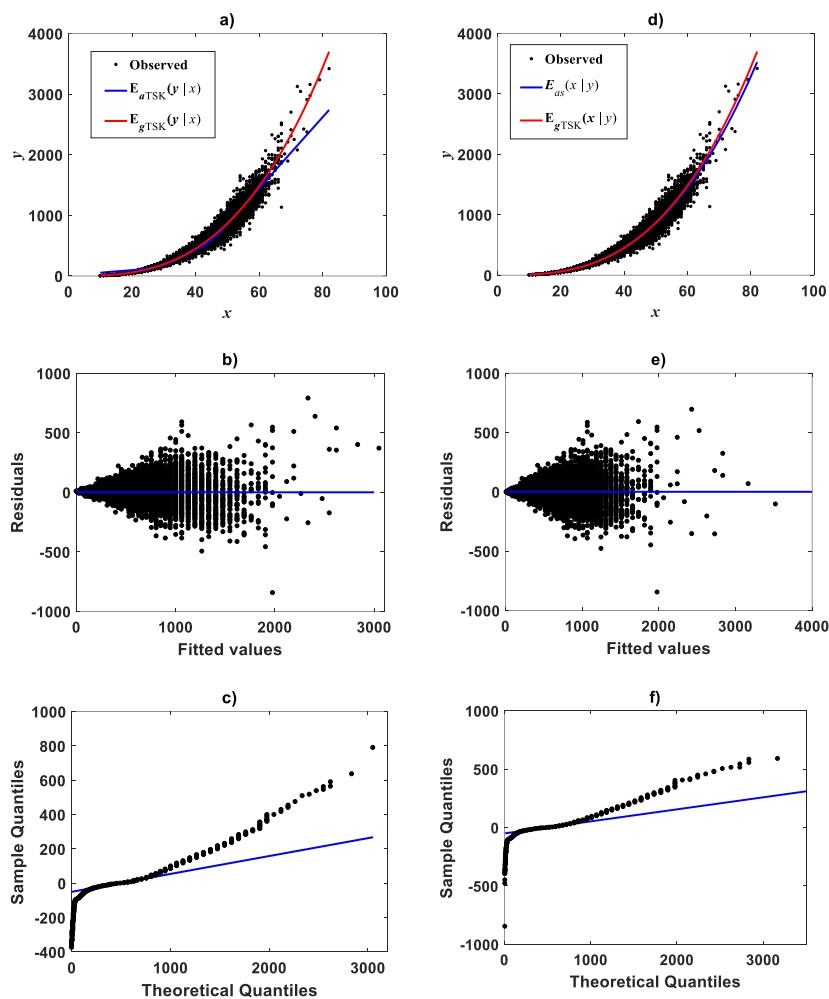

**Figure 15  TSK-MPCA model identified on the *De Robertis & Williams (2008)* data.** (A) shows spread about the interpolation mean response $E_{aTSK}(y|x)$ produced by a $r_a = 0.22$ and $q = 3$ fit of the TSK-MPCA fuzzy model. Plot exhibits a greater adequacy of $E_{gTSK}(y|x)$ obtained by retransformation of output of TSK-PLA fitted to this data. (B) and (C) residual spread and QQ plots for $E_{aTSK}(y|x)$ one to one. (D) presents spread about the mean response $E_{as}(y|x)$ produced by a fit of Huxley's formula of simple allometry compared to $E_{gTSK}(y|x)$. We can be aware that $E_{gTSK}(y|x)$ grants similar reproducibility features. (E) and (F) residual spread and QQ plots accompanying $E_{as}(y|x)$.

**Table 9  Model performance metrics for TSK-PLA and TSK-MPCA models fitted on the *Mascaro et al. (2011)* data set.** Included metrics are: AIC, CCC, $R^2$, SEE, MPE, and MPSE.

| Method | $r_a$ | $q$ | AIC | $\rho_c$ | $R^2$ | SEE | MPE | MPSE |
|---|---|---|---|---|---|---|---|---|
| $E_{aTSK}(y|x)$ | 0.855 | 2 | 305.32 | 0.9785 | 0.9579 | 35.09 | 15.7416 | 33.14 |
| $E_{aTSK}(y|x)$ | 0.52 | 3 | 308.62 | 0.9749 | 0.9530 | 37.0804 | 16.6307 | 111.49 |
| $E_{gTSK}(y|x)$ | 0.80 | 2 | 314.21 | 0.9720 | 0.9434 | 40.7013 | 18.2547 | 22.31 |

# DISCUSSION

A logarithmic transformation in allometry is often vindicated as a natural way to lodge a variation pattern resulting from multiplicative growth in plants and animals. Indeed,

*Gingerich (2000)* and *Kerkhoff & Enquist (2009)* state that a number of biological processes, (i.e., growth, reproduction, metabolism and perception), are essentially multiplicative and are therefore prone to fit in better to a geometric error model. Beyond biological arguments supporting the traditional approach, *Kerkhoff & Enquist (2009)* underline that fitting models to log-transformed data is seamlessly adequate, since taking into account proportional rather than absolute variation is more significant. Therefore, from this standpoint, the fact that log-transformation places numbers into a geometric domain could bestow advantages beyond a purely statistical convenience. Nevertheless, there are remarks that a logtransformation approach procedure produces biased results, and that direct nonlinear regression methods in arithmetical scale, should be preferred in parameter identification tasks (e.g., *Packard, 2013*; *Packard, 2009*; *Packard & Birchard, 2008*; *Packard & Boardman, 2008*). But, these views are debatable for a school of defenders of the traditional protocol. For instance, *Mascaro et al. (2014)*, stress on an important drawback in findings in *Packard (2013)* that refuted the traditional analysis method of allometry. This concerns the apparent significant bias linked to small values of the explanatory variable, that result from a use of nonlinear regression with the assumption of homoscedastic errors. Besides, *Mascaro et al. (2014)*, underline that a lack of a CF misled *Packard (2013)*, thereby explaining his assertion of biased results attributed to the logarithmic transformation protocol. Other practitioners have also placed a vigorous defense of this procedure, (e.g., *Ballantyne, 2013*; *Glazier, 2013*; *Lai et al., 2013*; *White et al., 2012*; *Xiao et al., 2011*). This is reasonably understood since inferences of many allometric studies could be invalidated by a substantiated rebuttal of this analysis method. But, *Packard (2017a)* asserts for instance, that adherence to a TAMA approach has been maintained even in situations when the resulting bivariate distribution is curvilinear in geometrical scale. Consequent pattern is generally referred as non-log linear allometry (*Packard, 2012b*; *Strauss & Huxley, 1993*; *Echavarría-Heras et al., 2019a*). Moreover, G.C. Packard has considered deviations from linearity in log-log plots of allometry as mainly attributable to a logtransformation itself (*Packard & Boardman, 2008*; *Packard, 2012a*; *Packard, 2012b*; *Packard, 2013*). From this perspective, overcoming the bias due to curvature in log scale requires extending complexity of Huxley's model of simple allometry in direct scales, which bears a paradigm of multiple parameter complex allometry (*Gould, 1966*; *Lovett & Felder, 1989*; *MacLeod, 2014*; *Bervian, Fontoura & Haimovici, 2006*; *Packard, 2012a*). Again, for promoters of the traditional approach this viewpoint sacrifices appreciation of biological theory in order to privilege statistical correctness (*Houle et al., 2011*; *Lemaître et al., 2015*; *Pélabon et al., 2018*). The approach underwent here demonstrates that a merging of points above can be achieved by evoking Huxley's report on the existence of a breakpoint in the log–log plot of chela mass vs. body mass of fiddler crabs (*Uca pugnax*) (*Huxley, 1924*; *Huxley, 1927*; *Huxley, 1932*). A generalization of this perspective explains adoption of a polyphasic loglinear allometry paradigm (*Packard, 2016*; *Gerber, Eble & Neige, 2008*; *Strauss & Huxley, 1993*; *Hartnoll, 1978*). This notion bestows curvature in geometrical space as determined by breakpoints interpreted as thresholds for transition among successive growth phases. Formally, this conception adds complexity for improving statistical consistency while keeping the meanings of allometric exponents as Huxley's original formulation.

Table 10 **Model performance metrics for TSK-PLA and TSK-MPCA models fitted on the *Huxley (1932)* Uca pugnax data set.** Included metrics are: AIC, CCC, $R^2$, SEE, MPE and MPSE.

| Method | $r_a$ | $q$ | AIC | $\rho_c$ | $R^2$ | SEE | MPE | MPSE |
|---|---|---|---|---|---|---|---|---|
| $E_{a\text{TSK}}(y|x)$ | 0.668 | 2 | 232.81 | 0.9986 | 0.9972 | 22.42 | 2.52 | 32.35 |
| $E_{g\text{TSK}}(y|x)$ | 0.8 | 2 | 199.72 | 0.9996 | 0.9993 | 11.56 | 1.30 | 1.8548 |
| $E_{g\text{TSK}}(y|x)$ | 0.2 | 7 | 160.43 | 0.9999 | 0.9998 | 5.27 | 0.59 | 0.9577 |

Table 11 **Model performance metrics for TSK-PLA and TSK-MPCA models fitted on the *De Robertis & Williams (2008)* data set.** Included metrics are: AIC, CCC, $R^2$, SEE, MPE and MPSE.

| Method | $r_a$ | $q$ | AIC | $\rho_c$ | $R^2$ | SEE | MPE | MPSE |
|---|---|---|---|---|---|---|---|---|
| $E_{as}(y|x)$ | — | — | 324,010 | 0.9812 | 0.9630 | 60.2367 | 0.1401 | 7.3767 |
| $E_{a\text{TSK}}(y|x)$ | 0.22 | 3 | 330,531 | 0.9762 | 0.9539 | 67.3087 | 0.1565 | 11.6985 |
| $E_{g\text{TSK}}(y|x)$ | 0.50 | 1 | 324,613 | 0.9812 | 0.9622 | 60.0856 | 0.1415 | 7.2992 |

Conventional approaches have handled curvature in geometrical space by means of polynomial regression (*Kolokotrones et al., 2010*; *Lemaître et al., 2014*; *MacLeod, 2010*; *Glazier, Powell & Deptola, 2013*; *Tidière et al., 2017*; *Echavarría-Heras et al., 2019a*). Nevertheless, complexity underneath precludes accounting for heterogeneity as determined by break-point allometry. Conventional identification procedures also offer refined broken-line regression protocols (*Beckman & Cook, 1979*; *Ertel & Fowlkes, 1976*; *Muggeo, 2003*; *Tsuboi et al., 2018*; *Ramírez-Ramírez et al., 2019*; *Echavarría-Heras et al., 2019a*). Nevertheless, this slant relies on nonlinear regression that requires starting values for break-point estimation. Besides, crucial profile log likelihood could be log-concave so local maxima problems may exist. Surpassing this inconvenience may depend on using smoothed scatter plots to get candidate break points and consider several additional trials for estimation sensitivity to different starting points. Also necessary inferences on estimates could make implementation difficult (*Julious, 2001*; *Muggeo, 2003*).

The approaches in *Bitar, Campos & Freitas (2016)*, *Echavarria-Heras et al. (2018a)* and in *Echavarria-Heras et al. (2019b)* typify fuzzy logic based hybrid paradigms aimed to allometric examination. Present TSK constructs can be placed in this framework. Moreover, as our results demonstrate offered fuzzy paradigm can naturally host complexity as intended in a break point assimilation of allometry. Moreover, conceived TSK arrangements offer direct-intuitive and starting value free identification of breakpoints. Certainly, beak points as conceived here correspond to points of intersection of TSK-firing strength factors. Besides, intervals in between break points can be interpreted as dominance realms of corresponding sub models. The TSK break point identification in geometrical space for the *Echavarría-Heras et al. (2019a)* and the *Mascaro et al. (2011)* was paralleled by conventional broken-line regression. This confirms consistent capabilities by the fuzzy paradigm to identify heterogeneity in of the logtransformmed response. Thus, the offered TSK fuzzy model can be considered a tool entailing efficient automatic detection of weighted polyphasic log linear allometry patterns. And, the fact that the TSK model identified linearity in geometrical space for the *De Robertis & Williams (2008)* data demonstrates this

approach can adapt complexity as necessary in an efficient way. But, we must emphasize that consistency of results in arithmetical space hinges on suitability of CF form. We suggest contemplating the optimal reproducibility criterion around Eq. (10) for this matter.

Motivation for the present research mainly stirred from the idea that identification based on a logarithmic transformation is suited for allometric examination. Visual inspection of TSK proxies fitted in geometrical space, as well as, included model performance metrics provides partial validation of our assertion. But, from the perspective of MPCA proponents, validation of detected heterogeneity should be made on the original arithmetic scales. Moreover, the addressed TSK-MPCA proxy corresponds to an expression of the general output of the TSK fuzzy model involving linear consequents in arithmetical scales. This arrangement is consistent with a MPCA approach as conceived in *Lovett & Felder (1989)*. Furthermore, identification of a TSK-MPCA arrangement allows examination of break point allometry in arithmetical scales. Existence of break points in direct scales of data, confirms that a corresponding structure detected in geometrical space was not induced by effects of a logtransformation itself. And, using the Weierstrass approximation theorem, it can be demonstrated that the general output of a TSK fuzzy model can uniformly approximate any continuous function to arbitrarily high precision (*Ying, 1998*; *Zeng, Nai-Yao & Wen-Li, 2000*). Therefore, the high order interpolation capabilities of the TSK-MPCA scheme sets criterion to evaluate performance of retransformed TSK-PLA output $E_{gTSK}(y|x)$. Certainly, as our results demonstrate this can be achieved by comparing the reproducibility strength of $E_{gTSK}(y|x)$ against that of $E_{aTSK}(w|a)$ for a given data set. And in the present settings the offered TSK-PLA or TSK-MPCA approaches were equally suited. This demonstrates that it is possible to maintain a logtransformation as part of a consistent allometric examination arrangement. This is a controversial subject whose clarification seems to be overcome by adopting presently offered analytical arrangement.

*Packard (2012a)* applied conventional statistical methods to conclude that a break point in Huxley's *Uca pugnax* data (*Huxley, 1932*) was inexistent. Nevertheless, application of present fuzzy methods detected a break point shifted back from the locus Huxley conceived. Corroboration of existence of this point seems to endure a biologically meaningful interpretation by Huxley of existence of a threshold for a sudden change in relative growth of the chela at about the time crabs reach sexual maturity. Likewise, detected break point in *Zostera marina* could be interpreted as a threshold beyond which plant assigns to leaves a relatively greater amount of tissue to resist damage and separation from shoots induced by drag forces. This implies different scaling parameters among small and large leaves (*Echavarría-Heras et al., 2019a*; *Echavarría-Heras et al., 2018b*; *Echavarria-Heras et al., 2019b*). Similarly, a detected break point in Metrosideros polymorpha may suggest different allometric scaling depending on tree size. Certainly, resource allocation to different tree traits like diameter or height could vary through growth in response to different environmental-biotic settings, and also to changes in resource availability. In this perspective, a risk of suppression by competitors may drive small trees to assign more resources to increase height (*Echavarria-Heras et al., 2019b*). Then, past a threshold height at which suppression risk is at a minimum resource may be apportioned to horizontal growth parameters such as diameter, crown and root cover (*Weiner, 2004*; *Ramírez-Ramírez et al.,*

*2019*; *Echavarria-Heras et al., 2019b*). Therefore since the aim of allometric examination is understanding the biological processes that bring about covariance among traits, analytical approaches entailing break point identification must be preferred over conventional complex multi-parameter approaches (*Echavarria-Heras et al., 2019b*). Indeed, on spite of any gains in statistical fit attributable to the latter, characterization of inherent heterogeneity by the former could enhance biological insight. Particularly, a TSK-PLA slant could be a highly biologically significant model of allometry, because it can model the breakpoints while keeping the meanings of allometric exponents as Huxley's original formulation (*Echavarria-Heras et al., 2019b*).

As it is demonstrated by the steps in the derivation of Eq. (29), an imbedding of the TSK-PLA in the original theoretical perspective of allometry makes MPCA in arithmetical scale its logical consequence. By the same token the TSK-PLA approach grants direct–intuitive and starting values-free estimation of break points for transition among growth phases. We can also refer to benefits derived from the outstanding high order interpolation capabilities by this device. This functional mode of the TSK paradigm can be achieved by adjusting the value of the clustering parameter $r_a$ in Eq. (B7) (radii in supplied code) as to let the identification algorithm increase the number $q$ of interpolation sub models in Eq. (20) or Eq. (48). And, if we can manage to include a suitable CF form, we can assure a remarkable reproducibility strength of projections of values of the response in arithmetical scales. Nevertheless, unsuitable forms of membership functions could lead to inconveniences in the present TSK approach. Moreover, fitting results of the TSK-MPCA on the *Mascaro et al. (2014)* data exhibit the extent on what a combination of membership functions form and $r_a$ value can influence both break point detection and reproducibility strength (Table 9). We can be aware for instance that for membership functions in the form set by Eq. (51) consistent break point transference among geometrical and arithmetical scales is only achieved when $r_a = 0.52$ which implies heterogeneity set by $q = 3$. Nevertheless, this by the way leads to a penalization in reproducibility strength relative to a fit by $r_a = 0.855$. Setting a compromise between both fits depends on integration of previous knowledge into the analysis. This could help for instance by suggesting ad hoc forms of membership function with the aim of achieving high reproducibility and consistent break point placement relative to that previously estimated on geometrical space. In any event present digression on integration of subjective knowledge in the analysis of Huxley's data illustrates the extent on which a fuzzy logic approach can elucidate issues in allometric examination.

## CONCLUSIONS

The offered TSK-PLA as formalized by the $v_{TSK}(u, \boldsymbol{\pi})$ paradigm can be interpreted as a generalized tool for the analysis of log transformed allometric data, that allows to contemplate: (1) the regression arrangement of the TAMA way (the case $q = 1$ and $\vartheta^1(u) = 1$), (2) a generalized nonlinear model for identification of weighted polyphasic nonlinear allometry (the case $q > 1$). (3) A direct–intuitive identification of concomitant break points for transition among successive growth phases.

On spite of a seemingly complicated formal set up of the $v_{TSK}(u, \boldsymbol{\pi})$ scheme, this can be conveniently identified by loading logtransformmed data into the provided code. Analysis

of model performance metrics show that the mean response function $E_{gTSK}(y|x)$ deriving from retransformation of $v_{TSK}(u, \pi)$ to arithmetical space produces similar reproducibility strength as its counterpart $E_{aTSK}(y|x)$ following from identification of its arithmetical space TSK-MPCA counterpart $w_{TSK}(x, p)$. Available conventional like broken line or weighted linear segment mixture regression approaches could offer reasonable analytical paradigms. Nevertheless, the offered TSK approach bears a flexible computational assembly for previous knowledge integration in an intuitive-interactive way. The present digression on Huxley's break point illustrates this advantage in a more proper way.

Present results confirm the pertinence of the quotation of *Kerkhoff & Enquist (2009)*, on the uselessness of a distinction between logarithmic transformations and nonlinearity in many instances of allometric examination. Moreover, in our view, whenever we can manage to exhibit a suitable CF form proposed Takagi Sugeno Kang generalization can elucidate a glowing controversy. Surely, this paradigm allows the coexistence of the log transformation step claimed by practitioners as a must in allometry, and the unbiasedness of parameter estimates attributed to alternate direct nonlinear regression approaches in the original scale defended by many others.

However, the fact that TSK-PLA modeling provided meaningful interpretation in present settings does not rear this paradigm as a general tool of allometric examination. In the elucidating around Eq. (1) we established a condition on the response being positive and having a zero limit as covariate approaches zero. Therefore, the TSK-PLA slant essentially aims to analyse zero intercept allometries. And, there are instances where the initial timing of development of the trait itself and overall size are different. This situation will lead to consideration of a negative intercept in direct scales, ruling out transference of the examination into geometrical space. Then, modeling should be necessarily kept in direct scales and relying in MPCA turns out to be biologically reasonable. There are also situations where the error structure can be additive while the biological process underlying allometry is multiplicative. Again, this requests keeping the analyses on the arithmetic scales or modeling heteroscedastic errors in geometrical space. Certainly, we briefly addressed this approach while analyzing the eelgrass data. However, offering a heteroscedastic TSK-PLA protocol suited for the general settings requires further exploration.

## ACKNOWLEDGEMENTS

Professor G.C. Packard and two anonymous reviewers provided valuable comments and criticism that improved our final presentation. We thank A. De Robertis for data sharing.

## APPENDIX A. FORMAL ELUCIDATION OF THE $W_{TSK}(X,P)$ PROXY FOR $W(X,P)$

*Azeem, Hanmandlu & Ahmad (2000)* proposed the Generalized Fuzzy Model (GFM) from which the *Mamdani (1977)* & *Larsen (1980)* or the Takagi-Sugeno-Kang additive fuzzy models can be derived as particular cases. And *Gan, Hanmandlu & Tan (2005)* demonstrated that the conditional mean of a Gaussian Mixture Model and the defuzzified

output of a Generalized Fuzzy Model (GFM) are mathematically equivalent. From this it follows that the probability density function of a Gaussian mixture can be reduced to accommodate the TSK fuzzy model. Likewise, by using the Weierstrass approximation theorem, *Ying (1998)* demonstrated that the general output of a TSK fuzzy can uniformly approximate any continuous function to arbitrarily high precision. In what follows we describe steps in the derivation of a $w_{TSK}(x, \boldsymbol{p})$ proxy for $w(x, \boldsymbol{p})$ when the former is expressed as the general output of a first order TSK fuzzy model. In what follows, we review the steps in order to conceive a TSK surrogate $w_{TSK}(x, \boldsymbol{p})$ for $w(x, \boldsymbol{p})$. For that aim, we recall our definition of $w(x, \boldsymbol{p})$ as a continuous function $w(x, \boldsymbol{p}) : X \to Y$ with both domain and range as sets of real numbers and $\boldsymbol{p}$ a set of parameters, and such that the allometric response $y$ admits the representation $y = w(x, \boldsymbol{p})$.

In assembling a TSK fuzzy model, we consider a set $T_x$ containing a number $q$ of linguistic terms $\Phi_k$ associated to the input variable $x$ (*Dernoncourt, 2013*; *Mendel, 2001*; *Zadeh, 1989*; *Zadeh, 1972*; *Echavarría-Heras et al., 2018a*; *Echavarria-Heras et al., 2019b*). Namely

$$T_x = \left\{ \Phi_k \mid k = 1, 2, \ldots, q \right\}. \tag{A1}$$

Each linguistic term $\Phi_k$ is described by means a membership function $\mu_{\Phi_k}(x) : X \to [0, 1]$. Then, if $x_1, x_2, \ldots, x_n$ are the values that $x$ takes on, the characterization

$$A_k = \sum_{x \in X} \mu_{\Phi_k}(x)/x = \left\{ \mu_{\Phi_k}(x_1)/x_1, \;.., \mu_{\Phi_k}(x_n)/x_n \right\} \tag{A2}$$

is defined as a fuzzy set (*Dernoncourt, 2013*; *Mendel, 2001*; *Zadeh, 1965*).

In what follows, we use the symbol $L_x$ to denote the collection of membership functions describing the input variable $x$, that is,

$$L_x = \left\{ \mu_{\Phi_k}(x) \mid k = 1, 2, \ldots, q \right\}. \tag{A3}$$

$L_x$ is known as a fuzzy partition of the input variable $x$ in the domain $X$ (*Mendel, 2001*; *Bodjanova, 1993*; *Bezdek, 1981*).

In the same way, we designate a collection $T_y$ of linguistic terms $\Psi_j$ associating to the output variable $y$. Namely

$$T_y = \left\{ \Psi_j \mid j = 1, 2, \ldots, r \right\}. \tag{A4}$$

Likewise, each linguistic term $\Psi_j$ is described by means a membership function $\mu_{\Psi_j}(y) : Y \to [0, 1]$ such that, if $y_1, \ldots, y_m$ are the values that $y$ acquires (*Dernoncourt, 2013*; *Mendel, 2001*; *Zadeh, 1965*), then we can also consider the fuzzy set

$$B_j = \sum_{y \in Y} \mu_{\Psi_j}(y)/y = \left\{ \mu_{\Psi_j}(y_1)/y_1, \ldots, \mu_{\Psi_j}(y_m)/y_m \right\}. \tag{A5}$$

(*Mendel, 2001*). Concurrently, we have the collection $L_y$ of membership functions describing the output variable $y$

$$L_v = \left\{ \mu_{\Psi_j}(y) \mid j = 1, 2, \ldots, r \right\}. \tag{A6}$$

This way, the $L_y$ sets a fuzzy partition for the output variable $y$ in the domain $Y$ (*Mendel, 2001*; *Bodjanova, 1993*; *Bezdek, 1981*).

Additionally, for $i = 1, 2, \ldots, q$, we advance correspondences $i \rightarrow A_{k(i)}$ and $i \rightarrow B_{j(i)}$, therefore, we can contemplate antecedents $P^i(x)$ of the form

$$P^i(x) : [x \ is \ A_{k(i)}] \tag{A7}$$

and consequents $Q^i(y)$

$$Q^i(y) : [y \ is \ B_{j(i)}] \tag{A8}$$

backing inferential rules $R^i$

$$R^i : \left\{ \begin{matrix} if & : & P^i(y) \\ then & : & Q^i(y) \end{matrix} \right\}. \tag{A9}$$

We may now think about a single input-single output fuzzy inference system (*Mendel, 2001*). This is conceived as an application $F : X \rightarrow Y$ incorporating (1) a fuzzification module that characterizes the fuzzy partitions $L_x$ and $L_y$, (2) an inference engine that uses the rules $R = \bigcup_1^p \{R^i\}$ to convert a fuzzy input into a fuzzy output, and (3) a defuzzification operator $D$ that transforms the fuzzy set obtained by the inference engine into a crisp value $y$ in $Y$.

A first order single input-single output Takagi-Sugeno-Kang fuzzy inference system (*Sugeno & Kang, 1988*; *Takagi & Sugeno, 1985*) considers decision rules $R^i$ having an antecedent $P^i(x)$ of the form given by Eq. (A7) but with the consequent $Q^i(y)$ in expression (A8) taking a crisp functional form $f^i(x)$. That is, in the TSK fuzzy inference system we consider $R^i$ rules of the form

$$R^i : \left\{ \begin{matrix} if & : & x \ is \ A_{k(i)} \\ then & : & y = f^i(x) \end{matrix} \right\} \tag{A10}$$

for $i = 1, 2, \ldots, q$. We notice also that being the consequent a real number the use of a defuzzification operator is not necessary.

An important component in a TSK fuzzy model is the firing strength $\varphi^i(x)$ of the antecedent $P^i(x)$ of a rule $R^i$. For a first order single input-single output TSK fuzzy model we take

$$\varphi^i(x) = \mu_{\Phi_{k(i)}}(x). \tag{A11}$$

A normalized firing strength $\vartheta^i(x)$ takes a form (*Mendel, 2001*)

$$\vartheta^i(x) = \frac{\varphi^i(x)}{\sum_1^q \varphi^i(x)}. \tag{A12}$$

It follows that

$$\sum_1^q \vartheta^i(x) = 1. \tag{A13}$$

The final output $w_{TSK}(x, \boldsymbol{p})$ of the Takagi-Sugeno-Kang inference system is the normalized firing strength weighted average of all rule outputs (*Sugeno & Kang, 1988*; *Takagi & Sugeno, 1985*), that is,

$$w_{TSK}(x, \boldsymbol{p}) = \sum_{1}^{q} \vartheta^i(x) f^i(x). \tag{A14}$$

Where $\boldsymbol{p}$ stands for the set of parameters identifying the membership and consequent functions in Eqs. (A3) and (A10) one to one.

## APPENDIX B. IDENTIFICATION PROCEDURES FOR $W_{TSK}(X, P)$

Description of structure and parameter estimation of the TSK fuzzy model interrelate (*Echavarría-Heras et al., 2018a*). A first stage relies on Subtractive Clustering (*Castro et al., 2016*; *Chiu, 1994*). This technique sets decision rules $R^i$ and produces parameter estimates for the $\mu_{\Phi_k}(x)$ membership functions. This acquires as well the forms of the normalized firing strength factors $\vartheta^i(x)$. A second stage of the identification task is achieved by placing weight factors $\vartheta^i(x)$ in Eq. (A14) in order to obtain parameter estimates for the consequents of the rules $f^i(x)$. Regularly, this is achieved by means of recursive least squares methods (*Jang, Sun & Mizutani, 1997*; *Wang & Mendel, 1992*).

### The subtractive clustering method

The subtractive clustering method (SC) is an extension of the Mountain Function Clustering method (MFC) proposed originally by *Yager & Filev (1994)*. This procedure estimates cluster centers based on the notion of a density function. For clarifying aims, before we explain the SC procedure, it is convenient to describe the MFC method. Following *Yager & Filev (1994)* and *Chiu (1994)*, the MFC procedure assembles the following steps:

1. The set of $n$ data to be analyzed is arranged as a vector $X$ namely:
   $$X = \{x_1, x_2, \ldots, x_n\}. \tag{B1}$$

2. Generation a *nth* dimensional space grid on which the data is located. Intersections of grid lines provide nodes $N_i$. $i = 1, \ldots, m$. Cluster centers to be are restricted to grid nodes. The set of cluster centers to be denotes through $C$.

   Based on the distribution of the data devise a mountain function (MF). This represents a data density measure. The height of the mountain function at a node point $N_i$ is

   $$MF(N_i) = \sum_{k=1}^{n} e^{-\alpha d(x_k, N_i)}, i = 1, 2, \tag{B2}$$

   where $\alpha$ is a positive constant and $d(x_k, N_i)$ is a measure of the distance between data point $x_k$ and grid node $N_i$. Equation (B2) enunciates that the data density measure at a point $N_i$ is influenced by all data points $x_k$. Such a density measure varies inversely proportional to distance between data points and node $N_i$. The parameter $\alpha$ not only influences the maximum value of $MF(N_i)$ but also its smoothness.

   It can be ascertained that the values of the mountain function on nodes are closely dependent on density of data points in the neighborhood of $N_i$. Mountain function

values can be also interpreted as the potential suitability of a grid node to become a cluster center estimate. A node with many neighboring data points will have a large mountain function value. Cluster center $C_1$ is chosen as node $N_i$ such that $MF(N_i)$ attains the maximum value among all remaining nodes, formally

$$MF^1(C_1) = \max_i [MF(N_i)]. \tag{B3}$$

Since the nodes close to $C_1$ will also have high mountain-function values, it deems necessary to remove the effect of $C_1$ before obtaining the next cluster center $C_2$. A new mountain function $MF^2(N_i)$ is shaped by taking off a scaled Gaussian function centered at $C_1$ this eliminates the effect of the first cluster. Iteratively the mountain function, after eliminating the effects of the cluster center that was previously identified becomes

$$MF^{k+1}(C_{k+1}) = max\left[MF^k(N_i) - MF^k(C_k)\sum_{k=1}^n e^{-\beta d(C_k - N_i)}, 0\right]. \tag{B4}$$

3. The previous step is repeated until the number of desired cluster centers is found or until a stopping condition is met. This expresses in terms of the ratio of first maximum value of mountain function found $MF^1(C_1)$ to corresponding penultimate maximum value found $MF^{k-1}(C_{k-1})$. Iteration stops when this ratio attains a value less than a certain positive constant $\delta$,

$$\frac{MF^1(C_1)}{MF^{k-1}(C_{k-1})} < \delta \tag{B5}$$

Since, the mountain function has to be evaluated at each grid point the processing time of the mountain clustering method rises exponentially with dimension of task. A SC scheme amends these difficulties by taking data points as potential cluster centers. This implies processing time becoming proportional to problem size instead of problem dimension. Since this method does not take into account any grid intersections execution time is reduced. Nevertheless, the real clusters centers do not necessarily place at data points, but in most cases this approach offers a good approximation, to cluster center identification. The SC approach also bases on a function representing the distribution of the data. Actually, the SC method consists of equations very similar to those used in the MF method, steps in the later are:

1. The set of $n$ data to be analyzed is defined as follows:

$$X = \{x_1, x_2, \ldots, x_n\} \tag{B6}$$

2. Since, each data point is an aspirant for a cluster center a density measure at data point $x_i$ is defined as

$$D_i = \sum_{j=1}^n exp\left(-\frac{d(x_i, x_j)}{(r_a/2)^2}\right) \tag{B7}$$

where $r_a$ is a positive constant, this constant acts as the radius that defines the area of proximity to the potential cluster center. Once all density assessments are obtained, the one with the highest value is taken. Formally

$$C_1 = Max[D_i] \tag{B8}$$

this will identify a first cluster center $C_1$,

3. Next cluster center is acquired by subtracting a scaled Gaussian function centered at $C_1$, that is

$$D_i = D_i - C_1 exp\left(-\frac{d(x_i, C_1)}{(r_a/2)^2}\right) \qquad \text{(B9)}$$

where $r_b < r_a$, is a radius that defines the proximity area of the cluster center $C_1$.

4. The maximum value of rescaled densities $D_i$ of Eq. (B9) is chosen as cluster center $C_2$. This procedure is repeated until a desired number $m$ of cluster centers $C_1, C_2, \ldots C_m$ is determined.

## The recursive least squares method

The general least-squares problem establishes the output of a model $y$ as given by a linearly parameterized expression, namely

$$y = \gamma_1 h_1(\boldsymbol{x}) + \gamma_2 h_2(\boldsymbol{x}) + \cdots + \gamma_n h_n(\boldsymbol{x}), \qquad \text{(B10)}$$

where $\boldsymbol{x} = [x_1, \cdots, x_p]^T$ is the model's input values vector, $h_1(\boldsymbol{x}), \cdots, h_n(\boldsymbol{x})$ are known functions of $\boldsymbol{x}$, and $\gamma_1, \cdots, \gamma_n$ called regression coefficients are to be fitted.

Without loss of generality, we address the case $q = 2$ assuming consequent linear functions in the form given by Eq. (53), so that the general output of the TKS of Eq. (A14) is

$$w_{TSK}(x, \boldsymbol{p}) = p_1^1 \vartheta^1(x)x + p_1^2 \vartheta^2(x)x + p_2^1 \vartheta^1(x) + p_2^2 \vartheta^2(x). \qquad \text{(B11)}$$

with $\vartheta^i(x)$ given by Eq. (A12). Then, $w_{TSK}(x, \boldsymbol{p})$ as given by Eq. (B11) becomes a particular characterization of Eq. (B10) by taking model's input values $\boldsymbol{x} = [x]^T$, and $p_1^1, p_2^1, p_1^2$ and $p_2^2$ unknown parameters in the consequent functions.

To obtain parameter estimates, we take into account that in the present settings the target system to be modeled involves an input-output relationship $x \to w_{TSK}(x, \boldsymbol{p})$ being $x$ the descriptor variable and $w_{TSK}(x, \boldsymbol{p})$ standing for the response $y$. Therefore, we have a training data composing pairs $(x_k : y_k)$, for $k = 1, \cdots, m$ that stand for replicates of the considered input-output relationship. Therefore, in order to identify the unknown parameters $p_1^1, p_2^1, p_1^2$ and $p_2^2$, we must fill in for each data pair $(x_k : y_k)$, into Eq. (B11) in order to obtain the set of $m$ linear equations:

$$\left\{ \begin{array}{rcl} p_1^1 \vartheta^1(x_1)x_1 + p_1^2 \vartheta^2(x_1)x_1 + p_2^1 \vartheta^1(x_1) + p_2^2 \vartheta^2(x_1) & = & y_1 \\ \vartheta^1(x_2)p_1^1 x_2 + p_1^2 \vartheta^2(x_2)x_2 + p_2^1 \vartheta^1(x_2) + p_2^2 \vartheta^2(x_2) & = & y_2 \\ \vdots & \vdots & \vdots \\ \vartheta^1(x_m)p_1^1 x_m + p_1^2 \vartheta^2(x_m)x_m + p_2^1 \vartheta^1(x_m) + p_2^2 \vartheta^2(x_m) & = & y_m \end{array} \right\} \qquad \text{(B12)}$$

This system of equations can be equivalently written in a concise form $\boldsymbol{BP} = \boldsymbol{y}$, where $\boldsymbol{B}$ is the $m \times n$ matrix,

$$\boldsymbol{B} = \left\{ \begin{array}{llll} \vartheta^1(x_1)x_1 & \vartheta^1(x_1) & \vartheta^2(x_1)x_1 & \vartheta^2(x_1) \\ \vartheta^1(x_2)x_2 & \vartheta^1(x_2) & \vartheta^2(x_2)x_2 & \vartheta^2(x_2) \\ & & \vdots & \\ \vartheta^1(x_m)x_m & \vartheta^1(x_m) & \vartheta^2(x_m)x_m & \vartheta^2(x_m) \end{array} \right\} \qquad \text{(B13)}$$

$P$ the $n \times 1$ vector of unknown parameters,

$$P = \begin{Bmatrix} p_1^1 \\ p_2^1 \\ p_1^2 \\ p_2^2 \end{Bmatrix} \tag{B14}$$

and being $y$ the $m \times 1$ output values vector:

$$y = \begin{Bmatrix} y_1 \\ y_2 \\ \vdots \\ y_m \end{Bmatrix}. \tag{B15}$$

The i-th row of the data matrix $\begin{bmatrix} B \vdots y \end{bmatrix}$ is denoted by $\begin{bmatrix} b_i^T, y_i \end{bmatrix}$ and formally represented by,

$$b_i^T = \begin{bmatrix} \vartheta^1(x_i) x_i \vartheta^1(x_i) \vartheta^2(x_i) x_i \vartheta^2(x_i) \end{bmatrix}. \tag{B16}$$

Then, Eq. (B12) modifies to include an error vector $e$ that accounts for random noise or modeling error, that is,

$$y = BP + e. \tag{B17}$$

Since $e = y - BP$ then $e^T e = (y - BP)^T (y - BP)$, and if we let $E(P) = e^T e$ we will have

$$E(P) = \sum_{i=1}^{m} (y_i - b_i^T P)^2. \tag{B18}$$

We call $E(P)$ the sum of squared errors. Then, we need to search for a characterization $\hat{P}$ of the vector $P$, which minimizes $E(P)$. Furthermore, the vector $\hat{P}$ is known as the least-squares estimator (LSE) of $P$. Since $E(P)$ is in quadratic form, $\hat{P}$ is unique. It turns out that $\hat{P}$ satisfies the normal equation

$$B^T B \hat{P} = B^T y. \tag{B19}$$

Furthermore, $\hat{P}$ is given by

$$\hat{P} = (B^T B)^{-1} B^T y. \tag{B20}$$

A $n$-order least squares estimator $\hat{P}_n$ of $\hat{P}$ defined by means of the expression

$$\hat{P}_n = (B^T B)^{-1} B^T y \tag{B21}$$

is a description of $\hat{P}$ that associates to $n$ data pairs taken out of the training data set $(x_i : y_i)$. Once we have gotten $\hat{P}_n$ we can acquire the following estimator $\hat{P}_{n+1}$ with a minimum of effort, through a recursive least-squares estimator (RLSE) technique, a procedure where the $nth$ row of $\begin{bmatrix} B \vdots y \end{bmatrix}$, with $(1 \leq n \leq m)$ denoted by $\begin{bmatrix} b_n^T \vdots y_n \end{bmatrix}$ is recursively obtained. We now explain the procedure behind the RLSE method.

A new pair $\left(b_{n+1}^T; y_{n+1}\right)$ becomes available as the $(n+1)^{\text{th}}$ entry in the data set, producing the $\hat{P}_{n+1}$ estimate,

$$\hat{P}_{n+1} = \left(\begin{bmatrix} B \\ b_{n+1}^T \end{bmatrix}^T \begin{bmatrix} B \\ b_{n+1}^T \end{bmatrix}\right)^{-1} \begin{bmatrix} B \\ b_{n+1}^T \end{bmatrix}^T \begin{bmatrix} y \\ y_{n+1} \end{bmatrix}. \tag{B22}$$

Further, in order to simplify the notation, the pair $\left(b_{n+1}^T; y_{n+1}\right)$ will be symbolized by $\left(b^T; y\right)$ and we also introduce the $p \times p$ matrices $H_n$ and $H_{n+1}$ defined by means of

$$H_n = \left(B^T B\right)^{-1}, \tag{B23}$$

and

$$H_{n+1} = \left(\begin{bmatrix} B \\ b^T \end{bmatrix}^T \begin{bmatrix} B \\ b^T \end{bmatrix}\right)^{-1} \tag{B24}$$

or equivalently

$$H_{n+1} = \left(B^T B + bb^T\right)^{-1}.$$

Then $H_n$ and $H_{n+1}$ are related through

$$H_{n+1} = \left(H_n^{-1} + bb^T\right)^{-1}. \tag{B25}$$

Therefore, using $H_n$ from Eq. (B23) and $H_{n+1}$ from Eq. (B25), we explain why Eqs. (B21) and (B22) can be equivalently written in the form

$$\hat{P}_n = H_n B^T y \tag{B26}$$

and

$$\hat{P}_{n+1} = H_{n+1}\left(B^T y + by\right). \tag{B27}$$

From Eq. (B26) we have $B^T y = H_n^{-1}\hat{P}_n$, then replacing this result in Eq. (B27) we get

$$\hat{P}_{n+1} = H_{n+1}\left(H_n^{-1}\hat{P}_n + by\right). \tag{B28}$$

Now, from Eq. (B25) we have $H_n^{-1}\hat{P}_n = \left(H_{n+1}^{-1} - bb^T\right)\hat{P}_n$, so replacing this result in the above expression we get

$$\hat{P}_{n+1} = H_{n+1}\left[\left(H_{n+1}^{-1} - bb^T\right)\hat{P}_n + by\right],$$

then simplifying yields

$$\hat{P}_{n+1} = \hat{P}_n + H_{n+1}b\left(y - y^T\hat{P}_n\right). \tag{B29}$$

Thus $\hat{P}_{n+1}$ can be recursively identified in terms of the preceding estimate $\hat{P}_n$ and the new data pairs $\left(b^T; y\right)$. Furthermore, the current estimate $\hat{P}_{n+1}$ is expressed as the previous one $\hat{P}_n$ plus a correcting term based on the new data $\left(b^T; y\right)$; this adjusting term can be understood as an adaptation gain vector $H_{n+1}$ multiplied by a prediction error $\left(y - b^T\hat{P}_n\right)$ linked to the previous estimator $\hat{P}_n$.

Calculating $H_{n+1}$ as given by Eq. (B24) is computationally costly and requires the adaptation of a recursive formula. From Eq. (B25) we have

$$H_{n+1} = \left(H_n^{-1} + bb^T\right)^{-1}$$

Using the matrix inversion formulation of Lemma 5.6 in *Jang, Sun & Mizutani (1997)* with $A = H_n^{-1}$, $B = b$, and $C = b^T$, we obtain the successive recursive formula for $H_{n+1}$ in terms of $H_n$:

$$H_{n+1} = H_n - H_n b \left(I + b^T H_n b\right)^{-1} b^T H_n$$

equivalently,

$$H_{n+1} = H_n - \frac{H_n bb^T H_n}{I + b^T H_n b}. \tag{B30}$$

Summarizing, the recursive least-squares estimator for the problem of $AP + e = y$., where the $nth (1 \le n \le m)$ row of $\left[B \vdots y\right]$, denoted by $\left[b_n^T \vdots y_n\right]$, is sequentially obtained. It can be calculated as follows:

$$\begin{cases} \hat{P}_{n+1} = \hat{P}_n + H_{n+1} b_{n+1} \left(w_{n+1} - b_{n+1}^T \hat{P}_n\right) \\ H_{n+1} = H_n - \dfrac{H_n b_{n+1} b_{n+1}^T H_n}{I + b_{n+1}^T H_n b_{n+1}}. \end{cases} \tag{B31}$$

Notice that in stablishing this result we have recalled Eq. (B30) and the fact that we had previously set the convention that for easy of presentation, the pair $\left(b_{n+1}^T; y_{n+1}\right)$ would be symbolized by the expression $(b^T; y)$.

### genfis2.m + anfis.m training

Examination in geometrical space is achieved by the code: **main_fun_tsk_pla_model_fit**. Similarly, analysis in direct arithmetical scales relies on the: **main_fun_tsk_mpca_model_fit** counterpart, both based on a **genfis2.m** + **anfis.m** training (both functions are included into the code in the supplementary files section). As it is explained in Matlab user's manual for **genfi2.m** and **anfis.m** functions, given sets of input and output data, **genfis2** produces a Fuzzy Inference System (FIS). Output by **genfis2** suggest a primary FIS for **anfis.m** training. This is achieved through subtractive clustering. This is achieved by **genfis2.m** by means the **subclust.m** function that extracts a set of rules that model the behavior of the data. Once the antecedent membership functions are obtained the procedure uses RLS estimation to determine the consequent functions of each rule.

For the given input-output data the **genfis2.m** function produces a FIS of a TSK type. Moreover, the *XIN* and *XOUT* matrices yield one column per input and output of the FIS, respectively. **radii** (e.g., $r_a$ in Eq. (B7)) specifies the influence range or proximity area of the cluster center for each input and output dimension, assuming that the data falls within a hypercube unit (range [0 1]). Specifying a smaller cluster radius will generally produce smaller clusters in the data and, therefore, more rules. When **radii** is a scalar, it applies to all input and output dimensions. When **radii** is a vector it has an input for each input and output dimension (*Chiu, 1994*).

The descending gradient method is one of the oldest techniques to minimize, a given function defined in a multidimensional input space. This method bases many direct optimization methods for restricted and unrestricted problems. On spite of its slow convergence, its simplicity makes it the most used non-linear optimization technique. Formally

$$\boldsymbol{\varphi}_{next} = \boldsymbol{\varphi}_{now} - \eta\boldsymbol{G}. \tag{B32}$$

A slightly different formulation of Eq. (B32) results from a gradient normalization, that is

$$\boldsymbol{\varphi}_{next} = \boldsymbol{\varphi}_{now} - \kappa\frac{\boldsymbol{G}}{\|\boldsymbol{G}\|} \tag{B33}$$

being $\kappa$ the actual size of step ,which interprets as the Euclidean transition distance from $\boldsymbol{\varphi}_{now}$ to $\boldsymbol{\varphi}_{next}$, namely

$$\kappa = \|\boldsymbol{\varphi}_{next} - \boldsymbol{\varphi}_{now}\|. \tag{B34}$$

In order to typify (Eqs. B32) and (B33) the former one refers as simple descendent gradient and the later as its normalized version.

The term $\eta\boldsymbol{G}$ in Eq. (B32) stands for the extent of step. With a fixed $\eta$, step magnitude changes automatically in each iteration due to different gradients of $\boldsymbol{G}$. If the minimum point is on a flat surface or plateau, $\boldsymbol{G}$ tends to be infinitesimally small. Consequently, the simple descendent gradient in Eq. (B32) exhibits a slow convergence. On the other hand, for a fixed $\kappa$ the normalized simple descending gradient in Eq. (B33) always does the same steps, neglecting how steep the slope is *Chan & Fallside (1987)* and *Rumlhart, Hinton & Williams (1986)*. It is then necessary to actualize step size $\kappa$ for efficiency.

### Actualizing the $\kappa$ value

Adjusting $\kappa$ dynamically requires an adaptive strategy. Based on empirical observations, an initial step size $\kappa = 0.01$ can be updated according to the following couple of heuristic rules (*Jang, 1993*):

1. If the objective function (MRSE) undergoes m consecutive reductions, increase by $p$%.

$$\text{IF SSE}(\boldsymbol{\varphi}_{\text{next}}) < \text{SSE}(\boldsymbol{\varphi}_{\text{now}})\text{THEN}\kappa = \kappa * \kappa_{\text{inc}} \tag{B35}$$

2. If the objective function (MRSE) manifests $n$ consecutive combinations of an increase and a decrease, decrease by $q$%.

$$\text{IF MRSE}(\boldsymbol{\varphi}_{\text{next}})/\text{MRSE}(\boldsymbol{\varphi}_{\text{now}}) > 1.04(\text{maxSSEinc})\text{THEN}\kappa = \kappa * \kappa_{\text{dec}}. \tag{B36}$$

Representative values for m, : n, : p and q :are 4, 2, 10% ($\kappa_{inc} = 1.1$) and 10% ($\kappa_{dec} = 0.9$), respectively. These typical values are more or less arbitrarily chosen. This update strategy is incorporated into hybrid learning (**anfis.m**: descending gradient to update the parameters in the antecedents and RLS to update the parameters of the linear consequents).

Summarizing, the **radii** specifies the range of influence of the cluster centers (membership function centers (e.g., $\boldsymbol{\theta}$ vector in Eq. (21)) for each input and output dimension. Specifying a smaller cluster radius will generally produce smaller clusters in the

data and, therefore, more rules. The value of $\lambda$ in a membership function is related to the **radii** value and the range of entries through >

$$\lambda = (\mathbf{radii}.{}^\star Range(X))/ \operatorname{sqrt}(8.0) \tag{B37}$$

(*Chiu, 1994*). The $\kappa$ in the descendent gradient method adapts the $\boldsymbol{\theta}$ and $\boldsymbol{\lambda}$ values of the membership functions (cf. Eqs. (21) and (52)) in the antecedents of each iteration or epoch while minimizing the objective function.

# APPENDIX C. MODEL PERFORMANCE METRICS

Besides AIC and $\rho$ indices, model assessment in this examination relies on the SEE, MPE and MPSE indices based on statistics of squared and absolute deviations of observed to predicted values. According to *Parresol (1999)*, SEE, MPE and MPSE statistics as model performance metrics were first recommended by *Meyer (1938)*, then by *Schlaegen (1982)* and have subsequently been used by *Zeng & Tang (2011a)*; *Zeng & Tang (2011b)*. We provide ahead related formulae and explanation.

**Akaike information criterion (AIC)**

$$AIC = -2l\left(\hat{\theta}\right) + 2p. \tag{C1}$$

**Lin's Concordance Correlation Coefficient ($\rho_C$)**

$$\rho_C = \frac{2\rho\sigma_Y\sigma_X}{(\mu_X - \mu_Y)^2 + \sigma_Y^2 + \sigma_Y^2} \tag{C2}$$

with $\rho$ standing for Pearson's correlation coefficient. The $\rho_C$ index estimates through

$$\hat{\rho}_C = \frac{2S_{YX}}{(\bar{Y} - \bar{X})^2 + S_Y^2 + S_X^2} \tag{C3}$$

where

$$\bar{Y} = \frac{1}{n}\sum y_i, \bar{X} = \frac{1}{n}\sum x_i,$$

$$S_Y^2 = \frac{1}{n}\sum\left(y_i - \bar{Y}\right)^2, S_X^2 = \frac{1}{n}\sum\left(x_i - \bar{X}\right)^2,$$

$$S_{XY} = \frac{1}{n}\sum\left(x_i - \bar{X}\right)\left(y_i - \bar{Y}\right).$$

**Determination coefficient ($R^2$)**

$$R^2 = \frac{\sum\left(\hat{Y}_i - \bar{Y}\right)^2}{\sum\left(Y_i - \bar{Y}\right)^2} \tag{C4}$$

**Standard error of estimation (SEE)**

$$SEE = \sqrt{\sum\left(y_i - \hat{y}_i\right)^2/(n - p)} \tag{C5}$$

**Mean prediction error (MPE)**

$$MPE = t_\alpha(SEE/\bar{Y})/\sqrt{n} \times 100 \tag{C6}$$

**Mean percent standard error (MPSE)**

$$MPSE = \frac{1}{n}\sum \left| (y_i - \hat{y}_i)/\hat{y}_i \right| \times 100 \tag{C7}$$

AIC (*Akaike, 1974*) compares performance of candidate models. The model with lowermost AIC is considered the best among competitors. AIC sets a compromise between goodness of fit and complexity, expressing through log-likelihood and number of parameters this way penalizing inclusion of needless ones. AIC often interprets as an estimate of lost information when a model replaces the process generating the data. Lin's Concordance Correlation Coefficient ($\rho_C$) symbolized also by means of (CCC) measures the extent on what one variable (Y) reproduces another (X), that is, it represents a measure of the similarity (or agreement) between the two variables. CCC can be estimated, with sample sizes of at least ten pairs (x, y). The R square ($R^2$) also named determination coefficient interprets through the ratio (SS due to regression/Total SS corrected for the mean) and is mainly intended as a measure of closeness between response values and fitted linear regression models. R square takes values between zero and one and measures the proportion of the total variation of the response, around the average, explained by the model (*Echavarria-Heras et al., 2019b*). When $R^2$ attains its maximum value the response is fully explained by the predictors in the fitted linear regression model. According to *Parresol (1999)* using the coefficient of determination as a fit index aimed to compare performance of biomass models was firstly suggested by *Schlaegen (1982)*. Nevertheless, for nonlinear models a high $R^2$ value does not necessarily entails high reproducibility strength. SEE takes on non-negative values and is of extensive use in statistical texts and statistical software. It stands a comprehensive valuation of goodness of fit of a model to observed data, by measuring the accuracy of ($\hat{y}_i$) predictions gained from a fitted regression model. When SEE attains its minimum value, observed values of the response coincide with the fitted mean response function, that the model displays exact reproducibility of observed values (*Echavarria-Heras et al., 2019b*).

The MPE, which is signifying currently used to assess goodness of fit of a model, is a standardized version of the coefficient of variation $CV = (SEE/\bar{Y}) \times 100$ expressed as a percentage, as proposed by *Schlaegen (1982)*. MPSE bears a measure of the average absolute relative error, expressed as a percentage. The use of MPSE as a model assessment index was proposed by *Schlaegen (1982)*, but it had been previously suggested by *Meyer (1938)*, who regarded it as a measure of the absolute deviation of the expected and predicted responses, relative to the size of the prediction ($\left| y_i - \hat{y}_i \middle| \hat{y}_i \right.$) expressed as a percentage average (*Echavarria-Heras et al., 2019b*).

### Funding
The authors received no funding for this work.

### Competing Interests
The authors declare there are no competing interests.
## Author Contributions

- Hector A. Echavarria-Heras, Juan R. Castro-Rodriguez, Cecilia Leal-Ramirez and Enrique Villa-Diharce conceived and designed the experiments, performed the experiments, analyzed the data, prepared figures and/or tables, authored or reviewed drafts of the paper, approved the final draft.

## Data Availability

Data and code are available in the Supplementary Files.

Note on the De Robertis and Williams 2008 data: We acquired the De Robertis and Williams 2008 raw data directly from the authors. We agreed to use the data solely for fitting the TSK fuzzy model. We were not allowed to release the data to third parties. For access to the raw data, please contact Professor De Robertis: alex.derobertis@noaa.gob.

## Supplemental Information

Supplemental information for this article can be found online at http://dx.doi.org/10.7717/peerj.8173#supplemental-information.

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
