# Peer review of "Assessment of a Takagi–Sugeno-Kang fuzzy model assembly for examination of polyphasic loglinear allometry"

_PeerJ, doi:10.7717/peerj.8173_

## Round 0.1 · original submission · Major Revisions

This study addresses a controversial topic of what scale should be used in studies of allometry, which is also reflected in the reviews of the manuscript. While two reviewers thought the presented approach is of interest, the first reviewer had critical concerns regarding the transformation that was used. I would like to give the authors the opportunity to address these concerns and revise their manuscript accordingly. In particular, I'd advise the authors to address both the more technical comments regarding their methodology as well as clarifying the relationship with the original discussion on scaling in allometry.

·

Basic reporting

No comment.

Experimental design

No comment.

Validity of the findings

No comment.

Additional comments

The authors have formulated a rather complex protocol for converting problems of “complex allometry” into ones of “polyphasic, loglinear allometry.” Inasmuch as the new routine is based on logarithmic transformations, I see no real improvement over existing methods, all of which are flawed at some level or another by the transformation itself. If the authors are interested in the scaling relationship between, say, surface area vs. mass of eelgrass leaves, then these are the data that need to be examined graphically and then studied analytically --- not their logarithms. In both “complex allometry” and “polyphasic allometry,” the departure from linearity in log domain reflects on failure to identify the proper functional form for the allometric equation on the arithmetic scale. The present study does not address that problem.

Reviewer 2 ·

Basic reporting

This paper describes an interesting modeling approach. The topic is attractive. The theory is presented at the minimum understandable level and the application is trying to support the theory. The paper has potential to be appreciated by the readers and I suggest its improvement in terms of the general comments to the authors.

Experimental design

Please read the general comments to the authors.

Validity of the findings

Please read the general comments to the authors.

Additional comments

1) The main contribution must be specified.
2) The motivation of the approach must also be specified in the context of other similar modeling approaches. Authors' strong well-acknowledged results in the field should also be included in this discussion.
3) Discussion of related work on other nonlinear modeling approaches including fuzzy ones should be extended with the following papers, which recently came into my attention because they proved to be successful in various applications:
- A review on information accessing systems based on fuzzy linguistic modelling, International Journal of Computational Intelligence Systems, vol. 3, no. 4, pp. 420-437, 2010.
- New results in modelling derived from Bayesian filtering, Knowledge-Based Systems, vol. 23, no. 2, pp. 182-194, 2010.
- Predicting tDCS treatment outcomes of patients with major depressive disorder using automated EEG classification, Journal of Affective Disorders, vol. 208, pp. 597-603, 2017.
- Model-free sliding mode and fuzzy controllers for reverse osmosis desalination plants, International Journal of Artificial Intelligence, vol. 16, no. 2, pp. 208-222, 2018.
4) The paper contains some grammar problems. Their correction is needed.
5) Please arrange the modeling approach in terms of an identification algorithm with clear steps.
6) What about the convergence of the identification algorithm? Please discuss.
7) This is an application paper. It is not clear how the theory from the previous sections is applied in the application one. More details are necessary. The comment 5) helps.
8) Transitions from section to section should be smoother. The comments 3) and 7) help.
9) More details on the models are needed. How did you train them?

Reviewer 3 ·

Basic reporting

no comment. Some really minor issues are described in my comments to authors.

Experimental design

Some statistical tests are used without explanations. The "SC" method also does not have clear explanations. These methods should be explained in more detail prior to the manuscript being considered further.

Validity of the findings

no comment

Additional comments

Dear authors,

Here I completed reviewing a manuscript entitled "A Takagi-Sugeno-Kang fuzzy model assembly for examination of weighted polyphasic loglinear allometry", submitted for PeerJ. This study looks at a highly controversial topic of whether the original arithmetic scale or the geometric scale should be used in studies of allometry. Authors proposes the Takagi-Sugeno-Kang fuzzy model (hereafter referred to as TSK) as a universal model that includes both the fit of non-linear model on arithmetic scale and the fit of a linear model on geometric scale as special cases. Authors apply TSK for empirical data to demonstrate that the method identifies and estimates parameters of polyphasic loglinear allometry only when breakpoints are clearly visible. Based on these results, authors claim that TSK is a useful model in allometric studies that reconciles the heated debate on the use of log scale in allometry.

I like the general idea of this manuscript. Especially I agree with authors' view that a method that inclusively treats issues of scaling should eventually resolve the current division of opinions in the field. The presented method appears to be generally credible, and the validation of the method are conducted with full transparency (original data and source code are available). Results suggest that TSK performs well in all three data examined.

Overall, the presented manuscript would fill an important knowledge gap in studies of allometry and its relevant fields. However, there are a couple of shortcomings in the current manuscript. They should be given further treatments before the manuscript will be considered further.

---------------------------------- start

1. The biological context in studies of allometry

In the introduction, authors provides a concise overview of the issue of scaling in allometry. This has generally a good coverage of relevant papers and well written, but one important aspect is missing, which is about the biological context in studies of allometry. One of the most fundamental discrepancies between the proponents of multiple parameter complex allometry on arithmetic scale, namely Gary Packard and his collaborators, and supporters of traditional allometry (TAMA, in your manuscript) lies in the degree of appreciation of biological theory in place of statistical correctness (e.g. Lemaître et al. 2015 Biol. Lett. 11:20150144, Pélabon et al. 2018 Biol. J. Linn. Soc. 125:664-671). As I describe below in comments on the use of visual inspection, this study, being a totally legitimate and scientifically correct endeavor, still has a great possibility to nourish the "useless" divide between two schools of allometry researchers. The fundamental issue is that this study focuses on the argument that TSK provides a good statistical fit to the data, based on AIC, r2 etc. We can easily find models based on multi-parameter complex models of allometry on arithmetic scales that fits data equally well, or even better than TSK. Thus, if we are to judge the validity of models based on statistical fit alone, TSK may well not be the best model of allometry.

I thus suggest authors to be explicit that the appreciation of biological theory is one major axis of polarization between the two schools. Then, briefly argue that TSK could be a highly biologically meaningful model of allometry, because it can model the breakpoints while keeping the meanings of allometric exponents as Huxley's original formulation. This is the key part, in my view, that is lacking in the current manuscript. If authors are unfamiliar with these arguments, I suggest reading Lemaître et al. (2015), Pélabon et al. (2018) as a starting point. Also, pp. 19-20 of Houle et al. (2011, Q. Rev. Biol. 86:3-34) is an excellent material to read.

2. Similar existing models

Authors introduces TSK as a novel model in studies of allometry, which I certainly agree, but the current manuscript does not give credit to existing approaches to fit polyphasic loglinear allometry. For instance, various methods for "the broken-stick" or "the broken-line" regression are available (e.g. Muggeo, 2003 Statistics in Medicine 22:3055-3071) and applied in allometry studies (e.g. Tsuboi et al. Nat. Ecol. Evol. 2:1492-1500). These methods are trying to solve the very same issue that TSK incorporates. Authors therefore should 1) briefly review existing broken-stick models, and 2) explain what is the uniqueness of TSK, and what is specific benefit(s) of using TSK over existing models.

3. Visual inspection as part of the proof

Authors' argument on the validity of TSK in fitting empirical data is based on a combination of statistical support (AIC, r2, etc.) and visual inspection of data in log-log plot. Proponents of non-linear models in arithmetic scales would immediately claim that the visual inspection should be made on the original arithmetic scales, and in this scale, the breakpoints seen in the log scale will most likely disappear. As such, the argument will fall back to an unproductive loop that authors courageously try to solve.

My suggestion is to examine and discuss whether the breakpoints estimated by TSK could be understood in biologically meaningful manners. An important difference between complex multi-parameter allometry model and polyphasic loglinear allometry is the existence of breakpoints in the latter. Because the goal of allometry study is to understand the biological processes that generate covariance between traits, the polyphasic loglinear approach, including TSK, should be favored regardless of statistical fit if the breakpoint has important biological meanings.
* * *
Other comments

line 69: "In" spite of

Line 72-73: What is the exact unreliability of TAMA in this example? Please be specific.

Line 85-97: Is this a problem? If so, why?

Line 152: "Analyses"

Line 267 onwards: What is SC? This is not introduced anywhere earlier, but it seems to be very important step that analytically determines the number of breakpoints (q). This must be explained in details, either in Appendix or in the method section.

Line 272: "established"

Lines 352-374: in this paragraph, several statistical tests are presented without explanations of what they are. Let's take one example, "an Anderson-Darling test", in line 363-365. The value of A, as well as its associated p-value is absolutely meaningless without readers knowing what the test is, what the A is, and what the null hypothesis is. Thus, all of these tests here and elsewhere should be briefly explained in the method section.

Line 378: I guess you mean "b" not "d"

Line 382-383: the sentence ", while... heteroscedasticity" is redundant, so should be removed.

Line 488: Akaike's information "criterion"

Line 497: remove ", of Akaike's AIC index (Akaike, 1974)" to avoid redundancy

Line 594: "Kerkhoff"

Line 671: The term "glowing" seems to be incorrect here, but I am not sure what authors want to deliver in this sentence. Please read it again, and rewrite it accordingly.

---------------------------------- end

I hope my comments will help improving the manuscript further.

Sincerely yours,

---

## Round 0.2 · Minor Revisions

The authors have adequately addressed the more technical comments regarding their methodology and have also presented a detailed and convincing response to Gary Packard's concern that the logarithmic transformations results in a failure to identify the proper functional form. Given their response, I believe the manuscript merits publication and makes a valuable contribution to the field. The reviewers did provide additional minor comments that the authors should address before the manuscript will be accepted for publication.

Reviewer 2 ·

Basic reporting

I agree with the modifications done and the responses to my comments. The paper is improved and meets the level of publication in this journal. Therefore, I suggest again the publication of this paper.

Experimental design

I agree with the modifications done and the responses to my comments. The paper is improved and meets the level of publication in this journal. Therefore, I suggest again the publication of this paper.

Validity of the findings

I agree with the modifications done and the responses to my comments. The paper is improved and meets the level of publication in this journal. Therefore, I suggest again the publication of this paper.

Additional comments

I agree with the modifications done and the responses to my comments. The paper is improved and meets the level of publication in this journal. Therefore, I suggest again the publication of this paper.

Reviewer 3 ·

Basic reporting

no comment

Experimental design

no comment

Validity of the findings

no comment

Additional comments

Dear authors,

I deeply impressed by authors' serious effort on revising this manuscript in response to my and other reviewers comment. All my comments are satisfactorily responded. I only have a few minor suggestions that may further clarify your point. Please see attached file for detail of my suggestions.

Otherwise, I just want to leave one final comment on your manuscript. Although I fully see the value of your model and I agree with you that TSK model would be preferred over MPCA in many cases of biological research, there are some occasions in which MPCA can be biologically reasonable model. I can think of at least two.

First, MPCA can model a situation in which the initial timing of development of the trait itself and overall size are very different. This situation will generate negative intercept that cannot be modeled in the log-domain. Second, the error structure can be additive while the biological process underlying allometry is multiplicative. This situation also calls for analyses on the arithmetic scale or modeling of heterozchedastic errors in the log scale. These are two situations that I know of, but there might be potentially more of these. Empirically, these situations may not be very common in biology but they "can" exist. If I were to add anything to the manuscript further, I would make it explicit that MPCA can be biologically insightful.

Congratulations for your fine work. I am looking forward to seeing this study published.

Sincerely yours,

Annotated reviews are not available for download in order to protect the identity of reviewers who chose to remain anonymous.

---

## Round 0.3 · accepted · Accept

The authors adequately addressed the outstanding comments.